EMBO
Molecular Medicine

# Cell-free DNA methylome and fragmentome analysis for relapse monitoring of Ewing sarcoma

Sophie A Richardson [ID][1,5], Aram Safrastyan [ID][1,5], Mina Karimpour [ID][1,5], Gayatri Gulati [ID][1], Patrick J B Harker [ID][1], Alan Redfern [ID][1], Simon P Pearce [ID][1], Vsevolod J Makeev [ID][1], Bernadette Brennan [ID][2], Alexander T J Lee [ID][3], Alexandra Clipson [ID][1], Steven M Hill [ID][1], Caroline Dive [ID][1], Dominic G Rothwell [ID][1], Martin G McCabe [ID][3,4,6 ✉] & Florent Mouliere [ID][1,6 ✉]

## Abstract

Liquid biopsies and cell-free DNA (cfDNA) offer minimally invasive methods for the diagnosis and monitoring of Ewing Sarcoma (EwS). EwS have a low tumour mutational burden and their detection with plasma cfDNA is challenging. We hypothesised that analysing the cfDNA methylome and fragmentome could enhance sensitivity for detecting EwS and identifying disease recurrence. Using T7-MBD-seq, we conducted whole-genome and methylome sequencing of cfDNA from 87 serial samples of 23 patients with EwS and 3 patients with CIC-rearranged sarcoma (CIC). With EwingSign, a new machine learning model, we identified EwS or CIC in a test set for 11 out of 16 patients at diagnosis and 15 out of 18 clinically confirmed relapse events. 0 out of 24 non-cancer controls (NCC) were detected positive with EwingSign. When combined with global and regional fragmentome analysis, all 18 relapse cases were detected, with 15/18 detected by 2 or more modalities, and 1 out of 24 NCC was detected by one modality. These findings indicate that cfDNA methylome and fragmentome analysis, if validated in a larger cohort, could improve disease detection, monitoring and relapse identification in patients with EwS.

**Keywords** Cell-Free DNA; Ewing Sarcoma; Fragmentomics; Liquid Biopsy; Methylome
**Subject Categories** Cancer; Chromatin, Transcription & Genomics; Methods & Resources

## Introduction

Ewing sarcoma (EwS) is the second most common bone and soft-tissue cancer in children and young adults, albeit with an infrequent incidence of 1 case per 1.5 million globally (Grünewald et al, 2018). EwS is driven by a fusion between *FET* and *ETS* family genes, most commonly *EWSR1::FLI1* (Delattre et al, 1992). Disease relapse occurs in ~30–40% of patients with localised EwS and 60–80% of patients with metastatic disease (Stahl et al, 2011). Patients with EwS have a worse prognosis when relapse occurs within the first 2 years after diagnosis (Stahl et al, 2011), therefore it is imperative that patients are monitored for relapsed disease. The standard of care for monitoring of patients with EwS includes lung screening with chest X-ray and either CT or MRI scans to image the primary tumour site (Strauss et al, 2021). These methods are limited by their sensitivity for detecting small volume tumours, and their inability to be used for frequent, long-term monitoring, in part due to radiation exposure (Pearce et al, 2012). There is a clear, unmet need for a biomarker to detect relapse in patients with EwS in a readily implemented and cost-effective manner.

Liquid biopsies are bio-fluids collected from the body, most commonly blood, and are minimally invasive (Janssen et al, 2024). These bio-fluids contain cell-free DNA (cfDNA), cell-free RNA and other molecules that can be analysed with molecular and biochemical techniques (Song et al, 2022). Cancer cells also release DNA into the bloodstream, referred to as circulating tumour DNA (ctDNA) (Leon et al, 1977; van der Pol and Mouliere, 2019). Hence, the detection of both genetic and epigenetic features of cancer is possible with a liquid biopsy (Tivey et al, 2024). Due to its low mutational burden (Crompton et al, 2014; Brohl et al, 2014), the detection of genetic mutations in EwS can be challenging. Previous studies have detected the *EWSR1::FLI1* fusion in liquid biopsies using cfDNA (Krumbholz et al, 2016, 2021; Hayashi et al, 2016; Shukla et al, 2017; Shulman et al, 2018; Schmidkonz et al, 2020) or utilised the fragmentation patterns of cfDNA (Peneder et al, 2021). The *EWSR1::FLI1* fusion (Krumbholz et al, 2016; Schmidkonz et al, 2020), copy number aberrations (CNAs) (Peneder et al, 2021; Van Paemel et al, 2022; George et al, 2025) and cfDNA fragmentation (Peneder et al, 2021; George et al, 2025) have been monitored throughout diagnosis and relapse in cohorts of up to 95 patients with EwS. cfDNA methylation has been explored in liquid biopsies for the diagnosis of EwS (Van Paemel et al, 2021; Gelineau et al,

[1]Cancer Research UK National Biomarker Centre, University of Manchester, Manchester M20 4BX, UK. [2]Royal Manchester Children's Hospital, Oxford Road, Manchester M13 9WL, UK. [3]The Christie NHS Foundation Trust, Manchester M20 4BX, UK. [4]Division of Cancer Sciences, University of Manchester, Manchester, UK. [5]These authors contributed equally: Sophie A Richardson, Aram Safrastyan, Mina Karimpour. [6]These authors jointly supervised this work: Martin G McCabe, Florent Mouliere.
✉E-mail: martin.mccabe@manchester.ac.uk; florent.mouliere@cruk.manchester.ac.uk

2023), with evidence of distinct epigenetic features of EwS compared to other sarcomas apparent through tumour tissue analysis (Koelsche et al, 2018, 2021) as well as interpatient heterogeneity within EwS (Sheffield et al, 2017). cfDNA methylation has also been shown to improve the sensitivity of cancer detection compared to mutation-based analysis in the context of adult cancers (Lasseter et al, 2020) and has the capability to determine tissue of origin (Luo et al, 2021; Conway et al, 2024).

This pilot study aims to longitudinally profile cfDNA samples from patients with EwS and Capicua-rearranged sarcoma (CIC), two cancers with small round cell tumours driven by gene fusions, using the in-house T7-MBD-seq assay, an enrichment-based methylation capture method using a methyl binding domain protein (Conway et al, 2024; Chemi et al, 2022). T7-MBD-seq is an integrated workflow combining genome-wide methylation analysis and (shallow) whole genome sequencing (WGS) for the recovery of CNAs and cfDNA fragmentomic features. A machine learning classifier, EwingSign, was developed to detect EwS or CIC using cfDNA methylation features. Longitudinal analysis was performed to understand how ctDNA burden, cfDNA fragmentation and cfDNA methylation changed over time, following systemic treatment and with sequential episodes of disease relapse. Finally, individual features of the assay were compared to determine their performance for the detection of EwS- and CIC-associated signal at both diagnosis and relapse.

## Results

### Study cohort and design

In total, 90 plasma samples were collected from 26 patients with sarcomas: 23 with EwS and three with CIC (Fig. 1; Table EV1).

Diagnosis samples were collected prior to treatment for 16 patients (15 EwS and 1 CIC). For 13 of these patients, two plasma samples were collected during first-line chemotherapy (26 samples). For one of these patients, an additional sample was taken post-surgery. For four patients, samples were only collected during first-line chemotherapy (eight samples), with one of these patients also providing a post-surgery sample. For nine patients (six EwS and three CIC), plasma samples were collected at one or more disease relapse timepoints (18 samples), and in seven of these patients, liquid biopsies were also collected during treatment of relapsed disease (20 samples, Table EV2). The mean cfDNA concentration across all samples was 23.49 ng/mL of plasma (0.08–356.74 ng/mL of plasma, Fig. EV1A). The mean % cfDNA (50–700 bp) by TapeStation analysis was 85.56%, with 87/90 samples having >50% cfDNA (Fig. EV1B). Three first-line treatment samples were excluded from further analysis due to low cfDNA concentration (<1 ng/mL of plasma). The remaining 87 cfDNA samples were analysed with an in-house library preparation method with a bisulfite-free, enrichment-based approach for methylation and whole genome sequencing (T7-MBD-seq) (Chemi et al, 2022; Conway et al, 2024). An additional sample was excluded after T7-MBD-seq from further analysis due to failure of QC (hyperstable fraction <0.4). WGS was performed to a mean 1.71-fold coverage (0.3–5.5-fold coverage). About 107 non-cancer controls (NCCs) were analysed with T7-MBD-seq and split into cohort 1 ($n = 83$) and cohort 2 ($n = 24$, Fig. 1; Table EV3). Of the NCCs, 81 were previously published (Conway et al, 2024; Chemi et al, 2022) and 26 are new. 20/107 NCCs were age-matched to the patient cohort (<45 years). No batch effects were observed between the previously published and newly analysed samples. The proportion of ctDNA (ichorTF) was calculated based on the CNA profile of WGS (Fig. 2A; Table EV4) with ichorCNA (Adalsteinsson et al, 2017). The

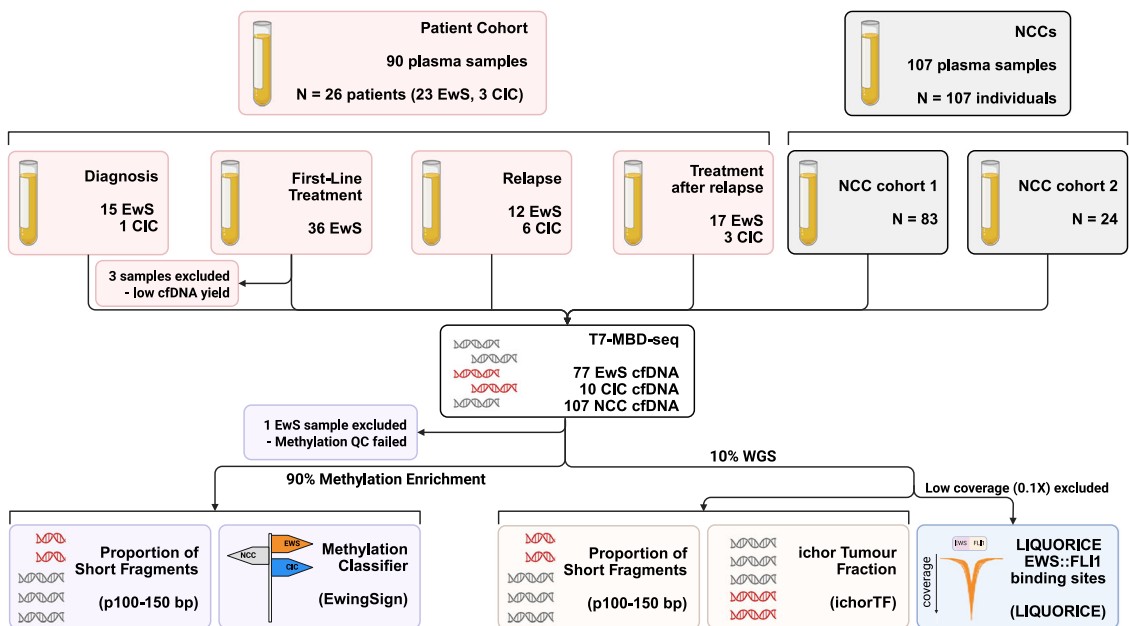

**Figure 1.   An overview of the study design.**

Study cohort flow chart depicting the number of patients and samples at each event, and the inclusion and exclusion of samples in cfDNA analysis. The analyses performed on T7-MBD-seq data are also shown. NCC: non-cancer control. EwS Ewing sarcoma, CIC CIC-rearranged sarcoma.

proportion of short cfDNA fragments (p100–150 bp) was determined both from the WGS and methylation enrichment data (Mouliere et al, 2018; Peneder et al, 2021). DELFI_FTK (Cristiano et al, 2019; Li et al, 2025) analysis was used to explore cfDNA fragmentation in a region-based manner, and LIQUORICE (Peneder et al, 2022) was used to explore cfDNA fragmentation within a set of EWS::FLI1 binding sites (Peneder et al, 2021). Methylation array data from EwS and CIC tissue samples (Koelsche et al, 2021) (Data Ref: Koelsche et al, 2021) and 83/107 NCCs (cohort 1) were used to train 25 ensemble XGBoost machine learning classifiers. Each of the 25 classifiers used an independent random split of NCC cohort 1 samples into a training set and a validation set, to mitigate potential bias arising from using a single split of NCC samples. The EwingSign classifier was obtained by averaging prediction scores across these 25 classifiers. The unseen test cfDNA data, including EwS, CIC and NCC cohort 2 samples, were subsequently used to evaluate the performance of EwingSign.

## Concurrent analysis of cell-free DNA methylation and fragmentomic analysis is the most sensitive for the detection of EwS and CIC disease

Using copy number aberrations (Fig. 2A), ctDNA was detectable in 9/16 diagnosis samples, 0/32 first-line treatment samples, 10/18 samples at disease relapse and 4/20 samples taken on treatment after relapse (cut-off of 0.03 ichorTF as advised by its authors (Adalsteinsson et al, 2017), Figs. 2B and EV2A). None of the 24 samples from NCCs were positive for ctDNA using ichorTF. ichorTF was significantly higher in samples from relapse events (median = 0.099) compared to first-line treatment samples (median = 0.005, $p < 0.05$, Mann–Whitney $U$-test).

Size distribution of cfDNA was calculated from coverage-normalised WGS and methylation enrichment data. Based on the global size difference between the NCCs and patients with EwS and CIC (Figs. 2C and EV2B,C, EV3A,B), and previous publications (Peneder et al, 2021; Mouliere et al, 2018), we chose the proportion of short fragments between 100 and 150 bp (p100–150 bp) to represent tumour-associated cfDNA. In the methylation enrichment data, p100–150 bp was deemed to be positive if the value exceeded the threshold of 0.1848 (highest p100–150 bp in NCC cohort 1; Fig. EV2B). 11/16 diagnosis samples, 4/32 first-line treatment samples, 15/18 relapse samples and 11/20 samples taken on treatment after relapse were positive with 1/24 NCC also above the threshold (Figs. 2D and EV2B). The p100–150 bp was significantly greater in diagnosis (median = 0.205) and relapse samples (median = 0.207) compared to first-line treatment (median = 0.167) and NCC samples (median = 0.155, $p < 0.05$, Mann–Whitney $U$-test). The p100–150 bp was also significantly greater in first-line treatment samples (median = 0.167) and treatment after relapse samples (median = 0.189) compared to NCC samples (median = 0.155, $p < 0.05$, Mann–Whitney $U$-test). Similar analysis of the WGS samples revealed fewer detected diagnosis (10/16) and relapse (11/18) cases, with all tested NCCs falling below the threshold of positivity (Fig. EV3C,D).

Region-based cfDNA fragmentation analysis was also performed to gain a deeper understanding of the observed global fragmentation shifts. To mitigate potential sequencing coverage-based confounding effects, coverage was normalised by down sampling both NCC and cancer WGS samples to ~1×. Subsequently, an

approach similar to DELFI (Cristiano et al, 2019) was used to assess the ratio of short to long fragments across the genome in 5 Mb bins (DELFI_FTK) (Li et al, 2025). This analysis suggested an enrichment in short fragments across chromosome arms 8p and 8q, which commonly have copy number gains in EwS (Tirode et al, 2014; Fig. 2A), in diagnosis and relapse samples compared to first-line treatment (Fig. EV4A). Additionally, unsupervised clustering of DELFI_FTK analysis showed separation of 9/16 diagnosis and 12/18 relapse events from NCCs (Fig. EV4B). Next, genomic region-set enrichment analysis was performed with LOLA (Sheffield and Bock, 2016), which indicated that significantly shorter fragments across 1 Mb bins from diagnosis and relapse EwS samples compared to NCCs were associated with EwS-specific open chromatin regions (Table EV5). Finally, LIQUORICE (Peneder et al, 2022) was used to examine the activity of the oncogenic transcription factor EWS::FLI1 by calculating the dip area in 982 EWS::FLI1 binding regions (Fig. 2Ei). 495 universally open DNase hypersensitive sites (DHS) were also analysed as a positive control (Fig. 2Eii). LIQUORICE was deemed to be positive (i.e., inferred activity of EWS::FLI1) if the dip area was less than −154.31 (lower bound of NCC prediction interval). 8/16 diagnosis samples, 12/18 relapse samples were positive using LIQUORICE, while 0/24 NCCs were positive (Figs. 2F and EV2C). In diagnosis samples (median = −207), the total dip area of EWS::FLI1 binding regions was significantly greater than first-line treatment samples (median = −50.7) and NCCs (median = 2.15, $p < 0.05$, Mann–Whitney $U$-test) (Fig. 2F). Similarly, the total dip area was also significantly greater in relapse samples (median = −281.29) compared to first-line treatment and NCC samples ($p < 0.05$, Mann–Whitney $U$-test) (Fig. 2F).

Our methylation-based machine learning model, EwingSign, classifies T7-MBD-seq cfDNA samples as either EwS, CIC or non-cancer. It comprises 25 ensemble XGBoost classifiers, each trained using a set of differentially methylated regions (DMRs) and an augmented training dataset generated from methylation array data for EwS and CIC tissue samples (Koelsche et al, 2021) (Data Ref: Koelsche et al, 2021), and T7-MBD-seq NCC cfDNA data from one of the NCC cohort 1 training set splits (Fig. 3A). In brief, for each of the 25 classifiers, a set of 200 DMRs were identified between EwS and NCC, CIC and NCC, and EwS and CIC samples (see Methods; Figs. EV5 and EV6). To mimic the low tumour content of cfDNA, an augmented training dataset of in silico mixture samples was then generated by mixing EwS tissue or CIC tissue methylation array data with one of the T7-MBD-seq NCC cohort 1 training sets, using a uniform distribution of tumour fractions (0.005–0.1). Each of the 25 ensemble classifiers comprised 100 sub-classifiers, with each sub-classifier trained on a subset of in silico mixture samples (see Methods). When applied to a test sample, each sub-classifier generated prediction scores for the EwS, CIC and NCC classes. The final prediction scores for the ensemble classifier were calculated as the mean prediction score for EwS, CIC and NCC classes across the 100 sub-classifiers. A sample was then deemed to be classified by EwingSign if a single class exceeded 0.5 mean prediction score, averaged across all 25 ensemble classifiers.

EwingSign was validated using in silico mixture samples of 28 independent EwS tissue samples from Patrizi et al, (2024) (Data Ref: Patrizi et al, 2024) mixed with 24 cohort 2 NCCs from the unseen test set. Each mixture sample was generated at each of 26 tumour fractions (range 0.001–0.05). The resulting classifier scores

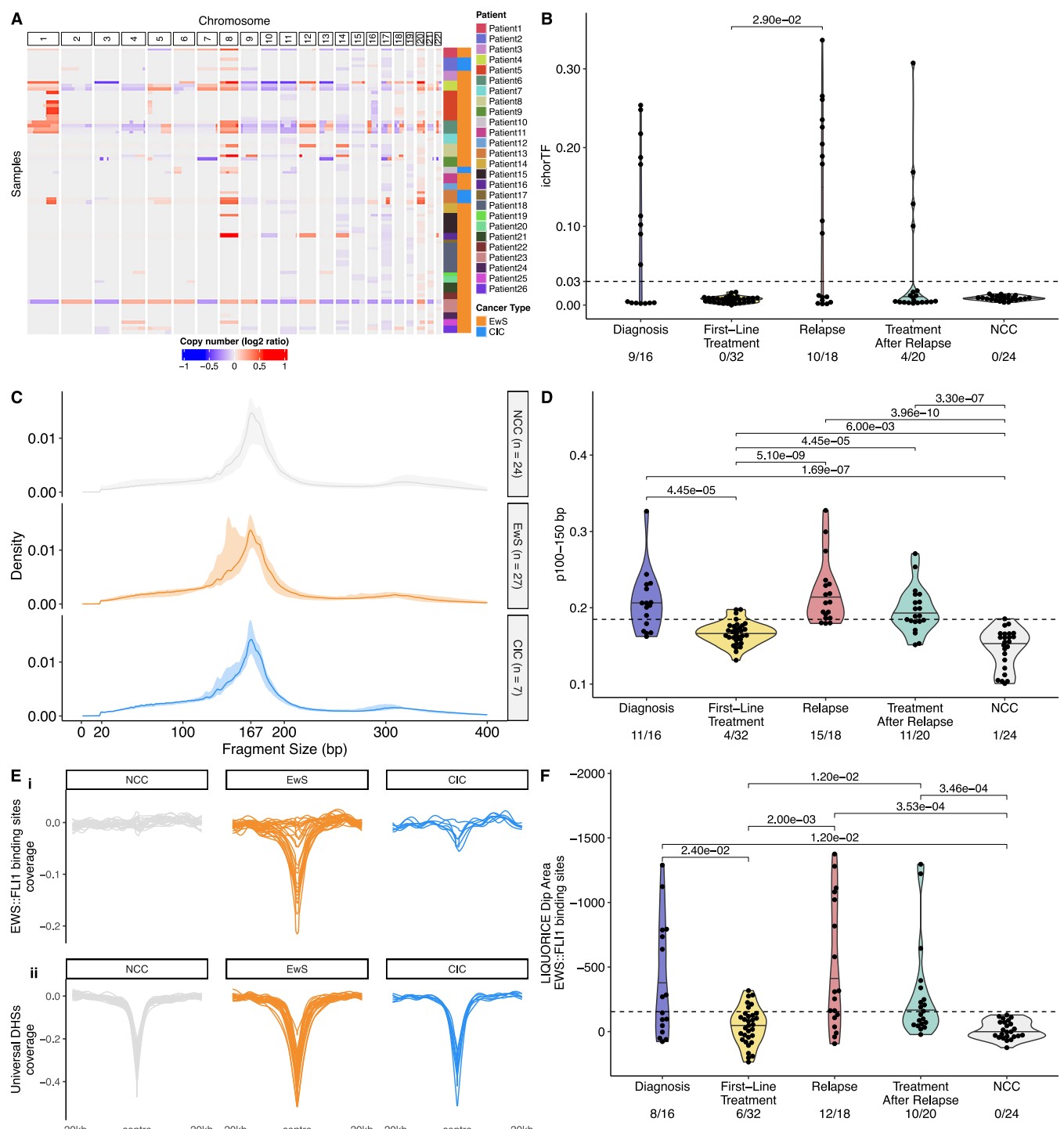

were then averaged across the NCCs. The mixture samples show EwingSign correctly predicts >75% of cases with a tumour fraction >0.013 (Fig. 3B).

EwingSign was applied to the unseen test set of EwS (23 patients and 76 samples), CIC (three patients and ten samples) and NCC (24 unseen samples) T7-MBD-seq cfDNA data. Prediction scores for EwS and CIC were combined to obtain a cancer prediction score, which achieved an area under the receiver-operating

characteristic (ROC) curve (AUC) of 0.960 for discriminating the diagnostic and relapse EwS and CIC samples from the NCC cohort 2 samples (Fig. 3C). In total, 11/16 diagnosis samples, 4/32 first-line treatment samples, 15/18 relapse samples and 11/20 samples taken on treatment after relapse were correctly classified as either belonging to the EWS or CIC class (Figs. 3D and EV7). One EwS sample and one CIC sample taken on treatment after relapse did not reach the cutoff for any class and therefore remained

**Figure 2. Fragmentomic analysis of EwS and CIC samples can be used for longitudinal analysis of patients' samples.**

(A) Heatmap of CNAs. Each row is a sample with patient and pathology annotated on the right. Each column is a region of the genome (1,000,000 bp windows) and separate chromosomes are denoted by the top annotation. Gains = red. Losses = blue. (B) Violin plots of ichorCNA tumour fraction at diagnosis, first-line treatment, relapse, treatment after relapse and in NCCs (NCC cohort 2). (C) Fragment size distribution for diagnosis/relapse cancer ($n = 34$) and NCC cohort 2 samples ($n = 24$). The line indicates the median size distribution for that group, and the ribbon indicates the range. NCC (grey); EwS (orange); CIC (blue). (D) Proportion of short fragments (100–150 bp) in methylation enrichment samples at diagnosis, first-line treatment, relapse, treatment after relapse and in NCCs (NCC cohort 2). (E) Coverage of EWS::FLI1 binding sites used to test for tumour-associated cfDNA (i) and universally open DNase hypersensitive sites (DHS) used as a positive control (ii) calculated using LIQUORICE. Plots are faceted to show NCC cohort 2 ($n = 24$ grey), diagnosis/relapse EwS ($n = 27$, orange) and diagnosis/relapse CIC ($n = 7$, blue) samples. (F) LIQUORICE analysis with 982 EWS::FLI1 binding sites at diagnosis, first-line treatment, relapse, treatment after relapse and in NCCs (NCC cohort 2). For all boxplots, the central line indicates the median, and the lower and upper boundaries of the box correspond to the 25th and 75th quartiles, respectively. The upper whisker extends to the largest value no further than 1.5 * the interquartile range (IQR), and the lower whisker to the smallest value no further than 1.5 * IQR. Each point indicates an individual sample. Two-sided Mann–Whitney U-test used for comparisons, only significant p values shown ($p < 0.05$). Source data are available online for this figure.

unclassified. All NCCs were correctly classified as NCC. In addition, comparison of EwingSign predictions with ichorTF shows that 19 EwS and CIC samples with undetectable ichorTF display high EwingSign cancer prediction scores, highlighting higher sensitivity of EwingSign compared to ichorCNA. (Fig. EV8). Detailed EwingSign prediction information for the unseen test set, across the 25 ensemble classifiers, is provided in Table EV6.

In summary, while p100–150 achieved a slightly higher AUC (0.974, Fig. 4A), EwingSign resulted in the greatest accuracy (86.2%) with 76.5% sensitivity for disease detection (26/34 diagnosis or relapse samples) and 100% specificity (0/24 NCCs). p100–150 was equivalent to EwingSign for the detection of relapse (15/18 incidences) and detected three relapse samples not identified by EwingSign, but had a lower specificity (95.8 vs. 100%, Table EV7). LIQUORICE detected disease in fewer samples compared to EwingSign (8/16 diagnosis, 12/18 relapse) but identified disease in 1 relapse sample, which was not detected by EwingSign, while maintaining a specificity of 100% (Fig. 4B). The combined use of EwingSign and p100–150 would identify all relapse samples (18/18), with however, a drop in specificity. Therefore, the most sensitive approach for disease detection without compromising specificity would be to use EwingSign and LIQUORICE concurrently (79.4% sensitivity and 100% specificity).

## Longitudinal liquid biopsy monitoring of EwS patients reflects disease dynamics

To detect relapse in a clinical setting, patients would be monitored longitudinally for disease with imaging techniques. We propose that liquid biopsies could augment this approach by providing additional molecular information. Here, we report the analysis of longitudinal blood samples from four exemplar patients with different disease trajectories and clinical outcomes (Fig. 5A–D). These were: Patient 1, whose disease did not relapse and responded to first-line treatment with all modalities supporting this. Patient 5, who responded to first-line treatment but later relapsed with all modalities detected at diagnosis, and all except ichorTF detected at relapse. Patient 8, who did not respond to first-line therapy and relapsed, with p100–150 bp potentially supporting this, but no signal observed in other modalities at first-line treatment. Patient 15, who responded to treatment of relapsed disease with all modalities supporting this.

Liquid biopsy samples from Patient 1 at diagnosis were positive by all four modalities (Fig. 5A–D). At the first on-treatment timepoint (day 91), LIQUORICE was positive, but all were below

the threshold for positivity at the final on-treatment timepoints at day 119.

Liquid biopsy samples from Patient 5 were positive for all four modalities at diagnosis, then below all assay thresholds except LIQUORICE and p100–150 bp throughout first-line treatment (Fig. 5A–D). At the first relapse timepoint (day 405), the sample was positive by p100–150 bp, LIQUORICE and EwingSign, but was negative by ichorTF. During treatment of this relapse and a second relapse, the samples continued to be positive for p100–150 bp, LIQUORICE and EwingSign, but ichorTF remained negative. Patient 5 then died 765 days after their diagnosis liquid biopsy.

Liquid biopsy samples from Patient 8 were positive by all four modalities at diagnosis (Fig. 5A–D). First-line treatment samples were negative for ichorTF, LIQUORICE and EwingSign. However, p100–150 bp increased above the threshold from day 46 to day 60. A further increase was observed in all modalities from day 60 to the clinical diagnosis of relapse at day 179, at which point EwingSign also classified the sample as EwS. The patient died 194 days after their first liquid biopsy.

Liquid biopsy samples from Patient 15 at relapse were positive by all four modalities. Five subsequent on-treatment samples at days 28, 48, 121, 141 and 162 were all below assay thresholds for ichorTF, LIQUORICE and EwingSign (Fig. 5A–D). p100–150 bp was positive only on day 141. Despite monitoring for 596 days there was no evidence of progressive disease either by liquid biopsy or clinically.

At least one modality was positive in all (18/18) relapse samples (Figs. 5E and EV9). All patients with relapsed disease had detectable disease by at least one modality at their first relapse incidence (9/9 patients). Of the patients that have not relapsed, 0/17 were positive by ichorTF, 1/17 was positive by p100–150 bp, 2/17 were positive by LIQUORICE and 2/17 were positive by EwingSign at their final timepoint. Only three patients with clinical relapse (patients 8, 12 and 5) have at least a pre-relapse plasma sample collected. Among them, all three of them have at least one cfDNA modality positive at clinical relapse. Patient 13 has no cfDNA sample collected between baseline and relapse (both are positives). Patient 8 has two samples collected on-treatment before relapse, one of them being positive for one modality. Patient 5 has two cfDNA samples collected before the first clinical relapse, one of them being positive for two modalities. The potential lead-on time is 119 days for patient 8, and 351 days for patient 5. These early data suggest EwS and CIC patients could have derived a benefit from liquid biopsies for the detection of relapse, but will require confirmation on larger cohorts.

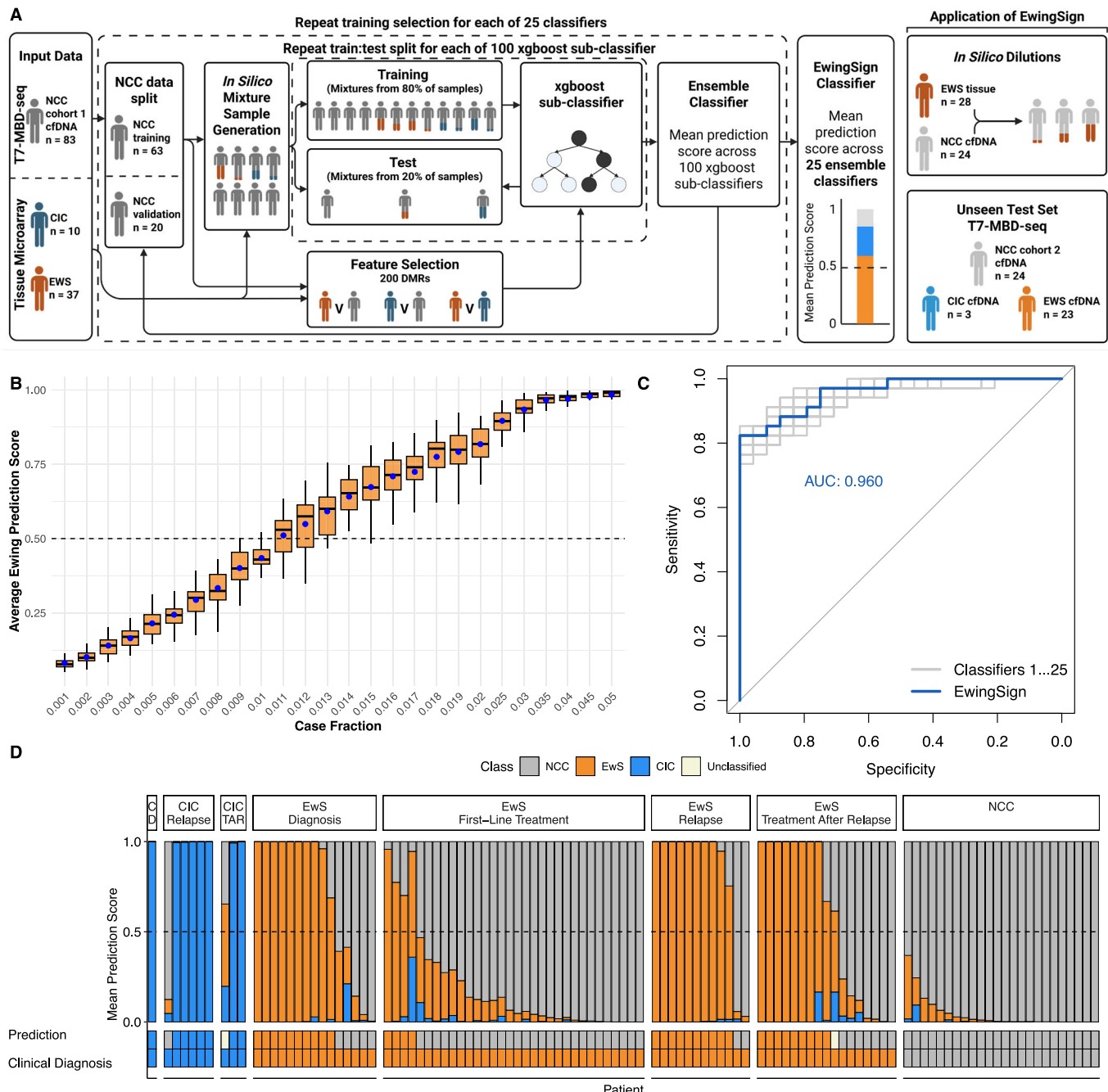

## Discussion

We sought to pilot a sensitive, high-throughput and cost-effective approach for detection of disease relapse of bone and soft-tissue sarcomas (EwS and CIC) via multi-modal analysis of cfDNA at baseline and in longitudinal blood samples using a single workflow, T7-MBD-seq. We show that using a cfDNA methylation machine learning classifier, EwingSign, provides a sensitive and specific detection of disease at both diagnosis and disease relapse. Disease was detected at diagnosis in 11/16 patients using EwingSign. In the five patients where no disease was detectable by liquid biopsy, patients all had localised disease. Using EwingSign, all patients with

relapsed disease had at least one positive relapse signal in their blood sample, with 83% of relapse events detected (15/18). LIQUORICE analysis of EWS::FLI1 binding sites resulted in the detection of 66.67% of relapse events (12/18). Using both Ewing-Sign and LIQUORICE concurrently results in the detection of 88.88% relapse events (16/18), suggesting that these patients could have potentially benefited from monitoring with a cfDNA-based liquid biopsy.

Previous studies of liquid biopsies in EwS have focused on the detection of *EWSR1::FLI1* fusion cfDNA molecules (Krumbholz et al, 2016, 2021; Hayashi et al, 2016; Shukla et al, 2017; Shulman et al, 2018; Schmidkonz et al, 2020), which are specific to EwS. Such

**Figure 3. A methylation classifier, EwingSign, is the most accurate modality for detection of EwS and CIC disease.**

(A) Graphical description of the EwingSign classifier comprising 25 ensemble XGBoost classifiers. For each ensemble classifier, the NCC cfDNA input samples (cohort 1) were split into training and validation subsets, and the training subset was used together with tumour tissue microarray data for feature selection and generation of in silico mixture samples. Mixture samples were subsequently divided into training and test sets, using an 80:20 split, to train 100 XGBoost sub-classifiers. An ensemble model was then constructed by averaging the prediction scores from these 100 sub-classifiers. The EwingSign prediction scores were then calculated as the mean prediction across the 25 ensemble classifiers. EwingSign was applied to in silico dilutions of EwS methylation data and unseen test cfDNA samples including NCC cohort 2, EwS and CIC samples. (B) Boxplots showing EwingSign EwS class prediction scores for in silico mixtures of EwS methylation data from 28 tissue samples (Patrizi et al, 2024, Data Ref: Patrizi et al, 2024) with 24 NCCs from NCC cohort 2. Mixtures were generated with tumour fractions ranging from 0.001 to 0.05 of EwS tissue methylation data. Each boxplot represents the distribution of 28 average EwingSign EwS prediction scores, averaged across all NCC mixtures for each EwS tissue sample. Boxplots indicate the interquartile range, horizontal lines denote medians, and blue points represent the mean prediction score for each tumour fraction. (C) Receiver-operating characteristic (ROC) curve and corresponding area under the curve (AUC) for the detection of (EwS or CIC) cancer signal by EwingSign, evaluated on the 34 diagnosis and relapse EwS and CIC samples, and 24 NCC samples. In addition to the ROC curve for EwingSign (blue line), also shown are ROC curves for each of the 25 individual ensemble classifiers that comprise EwingSign (grey lines). (D) EwingSign classifier results for unseen test samples. Each bar shows EwingSign prediction scores, averaged across 25 ensemble classifiers for an individual sample. The proportion of the bar fill colour indicates the EwS (orange), CIC (blue) and NCC (grey) scores, which all sum to a value of 1. Samples are faceted by cancer type first (CIC, EwS and NCC), then by timepoint (diagnosis, first-line treatment, relapse and treatment after relapse). Facet labels are abbreviated where necessary: C CIC, D diagnosis, TAR treatment after relapse. Source data are available online for this figure.

targeted approaches were shown to be less sensitive than a meta-learner using cfDNA fragmentation developed by Peneder et al, (2021). Therefore, we built upon the research of Peneder et al, (2021) to determine the utility of cfDNA fragmentation as a tool for monitoring EwS patients, although a limitation of this study is a lack of comparison between our methods and those for detecting molecular alterations or EwS fusion molecules. We observed high p100–150 bp fragments in a patient with no detectable ichorTF at either of their relapse timepoints. This was not unexpected, as it has been reported that only 63% of EwS cases have CNAs (Gillani et al, 2025), which are required for the detection of ctDNA using ichorCNA. In addition to high p100–150 bp fragment levels, patient 5 also had an indication of EWS::FLI1 activity by LIQUORICE and was classified as EwS by EwingSign, confirming that low ichorTF was associated with a lack of CNAs as opposed to a lack of tumour signal in this case. Additionally, we explored the dynamics of each assay modality and observed an increase in tumour-associated signal upon relapse, which has also been previously observed with ddPCR quantification of EWSR1::FLI1 fusion cfDNA molecules (Krumbholz et al, 2016).

As well as analysing the results of our EwingSign classifier longitudinally, we also observed the correct classification of 41/43 CIC and EwS samples where a cancer classification was made. The two samples that were not correctly classified were unclassified (no single class exceeded 0.5), and both were taken on treatment after relapse; timepoints where the disease would have been previously diagnosed. This implies that the differential methylation of sarcomas in tissue (Koelsche et al, 2021) may also be detectable in liquid biopsies using cfDNA methylation.

While the results presented here are promising, the following limitations apply: a small cohort, irregular sampling schedule, retrospective analyses, a non-risk-matched NCC cohort and limited genomic analyses. The small cohort size is due to the rarity of EwS (~ 2% of teenage and young adult cancer cases (Grünewald et al, 2018)) and this being a dual centre pilot study. For training EwingSign, we have mitigated the challenge of a limited training cohort by generating an augmented dataset using a cohort of tumour tissue data. Such an approach has previously been validated in the adult context (Conway et al, 2024). Further validation in external cohorts will be required to validate our observations and refine thresholds for the proposed features. The irregular, and opportunistic sampling schedule means we do not have the same sampling timepoints for every patient. Additionally, patients were not systematically monitored with liquid biopsies after initial treatment, relapse samples were only collected when disease was detected by standard of care monitoring, and all analysis was retrospective. Therefore, we could not validate whether liquid biopsies detect relapse earlier than the current standard of care. A clinical trial with a validated liquid biopsy assay for patient monitoring and decision-making would be required to determine whether earlier detection of relapse is possible, and to explore any benefits of earlier detection of relapse for patients with EwS. Currently, there is no standard of care for the treatment of relapsed EwS, however, there are clinical trials exploring various treatment options for the treatment of patients with relapsed EwS, such as rEECur (McCabe et al, 2022). Additionally, our NCC cohort (median age = 59.5 years) was not age- or risk-matched, and is significantly older ($p < 0.01$, Mann–Whitney $U$-test) than our patient cohort (median age = 21 years) due to the challenge of obtaining NCCs from a younger population. While across all age groups, NCCs showed significantly lower p100–150 bp than cancer patients (Fig. EV10A) and only a single age-matched NCC (<45 years) was assigned a cancer prediction score above the classification threshold by EwingSign (Fig. EV10B), further work is needed to fully examine the effects of age on EwS fragmentomic and methylomic analyses.

Here, we used epigenetic features of cfDNA for detection of and patient monitoring for a EwS or CIC sarcoma signal. However, there is also scope for these cfDNA-based assay features to be used to detect normal tissue toxicity and organ damage that may be caused by aggressive chemotherapy regimens used in paediatric patients with sarcoma (Peneder et al, 2021; Zhang and Li, 2023; Loy et al, 2024a). Greater sensitivity could be achieved by integrating cfDNA fragmentation and methylation into a single machine learning classifier. This was not possible within this study due to small sample numbers and, therefore lack of data to perform training and validation of a multi-modal machine learning classifier. Although cfDNA was analysed for four different modalities in our study, there are additional modalities such as single nucleotide variants (SNVs) and fusion events (Krumbholz et al, 2016, 2021; Hayashi et al, 2016; Shukla et al, 2017; Shulman et al, 2018) that could further improve a multi-modal liquid biopsy for EwS; for example, detection of STAG2 mutations may be prognostic (Shulman et al, 2022). Moreover, other analytes, such as

cfRNA, may improve sensitivity (Tao et al, 2023) and inform on toxicity (Loy et al, 2024b).

In conclusion, this study adds to the previous publications on liquid biopsies that begin to show promise for monitoring of patients with EwS and detection of relapse events. Validation in larger cohorts is warranted to determine the optimal combination of blood-based analytes (cfDNA features and other approaches) and their clinical utility, embracing a minimally invasive approach to augment imaging for children and young adults with EwS.

# Methods

### Reagents and tools table

| Reagent/resource | Reference or source | Identifier or catalogue number |
| --- | --- | --- |
| **Oligonucleotides and other sequence-based reagents** | | |
| T7dT_U1_Lcf oligo (sample barcode) | Integrated DNA Technologies | N/A |
| T7_Prim1 oligo (100 nmol, HPLC purified) | Integrated DNA Technologies | N/A |
| 5′Adenyl-U2-C3sp oligo (100 nmol, HPLC purified) | Integrated DNA Technologies | N/A |
| RT_U2P_primer (100 µM) | Integrated DNA Technologies | N/A |
| IDT custom index PE1/PE2 | Integrated DNA Technologies | N/A |
| **Chemicals, enzymes and other reagents** | | |
| EZ1&2 ccfDNA kit | QIAGEN | 954854 |
| QIAamp MinElute ccfDNA kit | QIAGEN | 55284 |
| QIAsymphony DSP Circulating DNA Kit | QIAGEN | 937555 |
| TaqMan™ RNase P Detection Reagents Kit | Applied Biosystems | 4316831 |
| Sensifast Probe Hi-ROX Kit | Bioline (SLS) | BIO-82005 |
| TapeStation Cell-free DNA ScreenTapes | Agilent | 5067-5630 |
| TapeStation Cell-free DNA Reagents | Agilent | 5067-5631 |
| TE Buffer | Invitrogen | 12090015 |
| NEBNext Ultra II End-Repair/dA-Tailing Module | New England Biolabs | E7646L |
| FastAP Thermosensitive Phosphatase | Thermo Fisher | EF0654 |
| KAPA Hyperprep Kit, PCR-Free | Kapa Biosystems (Roche) | KK8503 / 7962355001 |
| Agencourt AMPure XP | Beckman Coulter | A63881 |
| DNA Methylation Control Package | Diagenode | C02040012 |
| NEB Epimark Methylated DNA Enrichment Kit | New England Biolabs | E2600S |
| Sensifast SYBR Hi-ROX Kit | Bioline (SLS) | BIO-92005 |
| HiScribe T7 High Yield RNA Synthesis Kit | New England Biolabs | E2040S |
| Agencourt AMPure XP | Beckman Coulter | A63881 |

| Reagent/resource | Reference or source | Identifier or catalogue number |
| --- | --- | --- |
| FastAP Thermosensitive Phosphatase | Thermo Fisher | EF0654 |
| T4 RNA Ligase 2, Truncated KQ | New England Biolabs | M0373L |
| KAPA HiFi HotStart PCR Kit | KAPA Biosystems (Roche) | KK2502 / 7958897001 |
| Invitrogen SuperScript IV Reverse Transcriptase | Thermo Fisher | 18090050 |
| KAPA Library Quantification Kit for Illumina Universal | KAPA Biosystems (Roche) | KK4906 / 7960417001 |
| Qubit RNA Broad Range (BR) Assay Kit | Invitrogen | Q102010 |
| Agarose | Invitrogen | 16500-500 |
| Tris-Acetate-EDTA (TAE) 50X Solution | Fisher | BP1332-1 |
| Gel Star Nucleic Acid Gel Stain | Lonza | 50535 |
| RNA Loading Dye (2X) | New England Biolabs | B0363S |
| ssRNA Ladder (50 bp) | New England Biolabs | N0364S |
| Gel Loading Dye (6X) | New England Biolabs | B7021S |
| DNA Standard Ladder (100 bp) | New England Biolabs | N3231S |
| Glycerol | Invitrogen | 15514-011 |
| NovaSeq 6000 S4 Reagent Kit v1.5 (200 cycles) | Illumina | 20028313 |
| NovaSeq 6000 S2 Reagent Kit v1.5 (200 cycles) | Illumina | 20028315 |
| PhiX Control v3 | Illumina | FC-110-3001 |
| **Software** | | |
| Umitools v1.1.5 | (Smith et al, 2017) | |
| Cutadapt v3.4 | (Martin, 2011) | |
| Bwa mem 0.7.18 | (Li, 2013) | |
| Samtools 1.16.1 | (Li et al, 2009) | |
| FastQC v0.11.9 | | |
| Qualimap v2.2.2 d | | |
| MultiQC v1.13 | | |
| QSEA v4.3.2 | (Lienhard et al, 2017) | |
| MESA v0.5.1 | https://github.com/cruk-mi/mesa | |
| Hmmcopy v1.44.0 | (Shah et al, 2006) | |
| ichorCNA v0.3.2 | (Adalsteinsson et al, 2017) | |
| NGSCheckMate v1.0.1 | (Lee et al, 2017) | |
| FinaleToolkit v0.10.7 | (Li et al, 2025) | |
| LOLA v1.30.0 | (Sheffield and Bock, 2016) | |
| LIQUORICE 0.5.6 | (Peneder et al, 2022) | |
| Nextflow v24.04.2 | (Ewels et al, 2020) | |
| **Other** | | |
| EZ2 Connect | QIAGEN | |
| QIAsymphony | QIAGEN | |
| Lightcycler 96 | Roche | |

| Reagent/resource | Reference or source | Identifier or catalogue number |
|---|---|---|
| TapeStation 2100 | Agilent | |
| Illumina NovaSeq 6000 | Illumina | |
| Streck cell-free DNA BCT | Illumina | 15073345 |

## Study design and sample collection

Patients with EwS or Ewing-like Sarcoma were recruited under the Manchester Cancer Research Centre (MCRC) Biobank EwS project (application 21_CADI_03) and consented to give samples for this study with appropriate ethical approval from the MCRC Biobank Tissue Bank Ethics (ref: 22/NW/0237). All patients were recruited at The Christie NHS Foundation Trust and the Royal Manchester Children's Hospital. The study was not blinded. Non-cancer control samples were collected from three sources, with informed consent: (1) The Community Lung Health Study (ethically approved study London—West London & GTAC Research Ethics Committee REC reference: 17/LO415); (2) The University of Manchester healthy normal volunteer study (University of Manchester Research Ethics Committee 4 (UREC4) approval no. 2017-2761-4606 and 2024-16142-38412); or (3) Purchased from Cambridge Bioscience (University of Manchester Research Ethics Committee approval no. 2019-7920-11797). The study conformed to the principles set out in the WMA Declaration of Helsinki and the Department of Health and Human Services Belmont Report.

## Blood sample collection

Up to 10 mL of whole blood was collected from patients in Streck DNA blood collection tubes (BCTs). Blood was spun at 2000×$g$ for 10 min at room temperature with the brake off. The plasma supernatant was collected and centrifuged again at 2000×$g$ for 10 min at room temperature. The resulting supernatant was collected, aliquoted and stored at −80 °C. Blood was processed after a minimum of 24 h at room temperature, and within 96 h of collection.

## cfDNA isolation and QC

cfDNA from patients with EwS or CIC were isolated from Streck DNA plasma (median = 4.5 mL, range = 1.5–6.5 mL) using the EZ1&2 ccfDNA Kit (QIAGEN) on the EZ2 Connect liquid automation platform (QIAGEN) using a custom protocol which used 8 mL plasma input and 45 µL elution volume. If Streck DNA plasma volume was less than 8 mL, PBS was used to supplement the volume up to 8 mL. For the NCC samples (not previously published), cfDNA was isolated from Streck DNA plasma using the QIAamp MinElute ccfDNA kit (QIAGEN) for all samples except NCC0112, which was isolated using the QIAsymphony DSP Circulating DNA kit (QIAGEN) on the QIAsymphony instrument (QIAGEN). All isolated cfDNA was quantified using the TaqMan™ RNase P Detection Reagents Kit (Applied Biosystems, Catalogue number: 4316831) with a standard curve (maximum of 1000 copies, then twofold down to a minimum of 15.625 copies). cfDNA quality was analysed using the TapeStation 2100 (Agilent) and the Cell-free

DNA ScreenTape Analysis (Agilent) according to the manufacturer's instructions.

One sample (patient 14, day 14) presenting with haemolysis underwent a bead-based size selection to preferentially remove high molecular weight material (>500 bp). Briefly, a 0.5X bead ratio was added to the sample diluted 1:4.4 in TE buffer (Thermo Fisher). The beads were separated from the solution with a magnet, and the eluate containing low molecular weight material was kept. The eluate then underwent a bead-based clean-up with a 2.5X bead ratio (AMPure XP Beads, Beckman Coulter) to remove salts that may interfere with downstream reactions.

## T7-MBD-seq library preparation

T7-MBD-seq was performed as described by Conway et al. (Conway et al, 2024) using cfDNA (input 5.25–25 ng). cfDNA samples underwent end-repair and dA-tailing (NEBNext® Ultra™ II End Repair/dA-Tailing Module, New England Biolabs), dephosphorylation (FastAP Thermosensitive Alkaline Phosphatase (1 U/µL), Fisher Scientific), then ligation to a sample-specific barcode overnight (Kapa HyperPrep Kit PCR-Free, Roche and custom oligo, Integrated DNA Technologies). After barcode ligation, samples were pooled (9 to 13 samples per pool, 112 to 254 ng per pool) and remaining adaptors removed via 0.8X bead clean-up (AMPure XP Beads, Beckman Coulter). 90% of the sample pool underwent a methylation enrichment with the EpiMark® Methylated DNA Enrichment Kit (New England Biolabs), with the remaining 10% of the library pool retained for WGS. Both the enriched and unenriched sample pools were amplified by HiScribe T7 High Yield RNA Synthesis Kit (New England Biolabs) overnight to produce amplified RNA (aRNA). Methylation enriched aRNA underwent a single 1X bead clean-up and WGS aRNA underwent two 1X bead clean-ups prior to quality control checks using agarose gel electrophoresis and Qubit RNA Broad Range assay (Thermo Fisher Scientific). aRNA was dephosphorylated (FastAP Thermosensitive Alkaline Phosphatase (1 U/uL), Fisher Scientific), then ligated to an adapter containing an Illumina read 2 sequencing primer-compatible sequence (Custom oligonucleotide, Integrated DNA Technologies) overnight. Ligated aRNA underwent a 1X bead clean-up prior to reverse transcription (Superscript IV Reverse Transcriptase, ThermoFisher Scientific). A 1.5X bead clean-up was performed prior to PCR (KAPA HiFi PCR Kit HotStart PCR Kit, with dNTPs 250U, Roche) with unique dual index primers. A final 1.8X bead clean-up was performed prior to library quantification by qPCR (KAPA Library Quantification Kit for Illumina Universal, Roche). The enriched and unenriched library pools were then pooled equimolar, and then together at a ratio of 2:3. This pool was sequenced 150 bp paired-end on an Illumina NovaSeq6000 with S4 or S2 reagents (Illumina) and 15% PhiX.

## Bioinformatic data analysis software

All data analysis was performed in R (v4.3.0).

## T7-MBD-seq analysis

T7-MBD-seq analysis was performed as described previously (Conway et al, 2024). Briefly, a Nextflow (24.04.2) (Ewels et al,

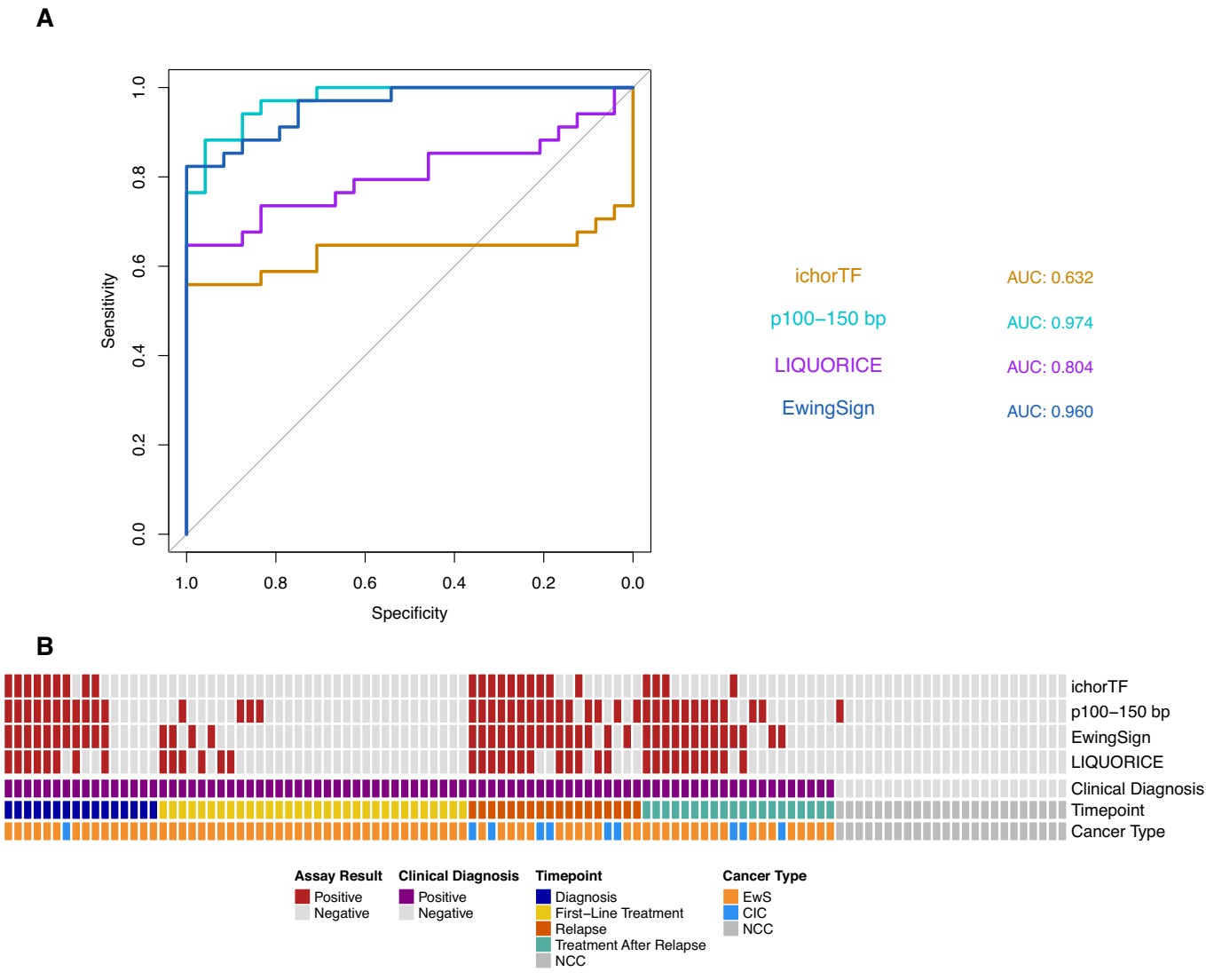

**Figure 4. Concurrent use of EwingSign and LIQUORICE achieves the highest cancer detection rate while maintaining 100% specificity.**

(A) Receiver-operating characteristic (ROC) curve and corresponding area under the curve (AUC) for the detection of (EwS or CIC) cancer signal using ichorTF (orange), p100–150 bp (turquoise), LIQUORICE (purple) and EwingSign (blue), evaluated on 34 diagnosis and relapse EwS and CIC samples, and 24 unseen NCC test samples. (B) A heatmap showing the frequency of positive (red) and negative (grey) assay results for each modality. From top to bottom, the rows show: ichorTF, p100–150 bp, EwingSign, LIQUORICE, clinical diagnosis, and timepoint and cancer type. Source data are available online for this figure.

2020) DSL2 pipeline (Conway et al, 2024) was used for FASTQ processing and generation of QSEA objects. Reads were trimmed to 91/61 basepairs (bp) for read 1 and read 2, respectively. The unique molecular identifier (UMI) was extracted using umitools (v1.1.5) (Smith et al, 2017), and samples were demultiplexed and adaptors trimmed with cutadapt (v3.4)(Martin, 2011). Reads were aligned to the reference genome (GRCh38) using bwa mem (0.7.18)(Li, 2013), deduplicated using both the R1 start position and UMIs with umitools (v1.1.5) and mate quality scores assigned with samtools fixmate (1.16.1)(Li et al, 2009). FastQC (v0.11.9), Qualimap (v2.2.2 d) and MultiQC (v1.13) were used throughout the pipeline for quality checks. The QSEA package (v4.3.2)(Lienhard et al, 2017) was used to analyse bam files alongside the R package MESA (v0.5.1, from https://github.com/cruk-mi/mesa). In brief, the

genome was tiled into 300 bp non-overlapping windows as previously described (Conway et al, 2024). Only paired reads where either end of the pair mapped with a Mapping Quality (MAPQ) score of at least 10, had a fragment length between 70 and 1000 bp and a distance along the reference genome of at least 30 bp were used in downstream analysis. Fragments were then uniquely assigned to windows according to the location of their midpoint. Copy Number Variations (CNV) were calculated for each sample from the non-enriched fraction, using hmmcopy (v1.44.0) (Shah et al, 2006) with base parameters over 1 Mbp windows. Normalised reads per million (NRPM) were generated using the CNV and the number of valid fragments in the sample, without applying trimmed mean of M values (TMM) normalisation. Beta-values (a scaled measure of methylation between 0 and 1) for each window in

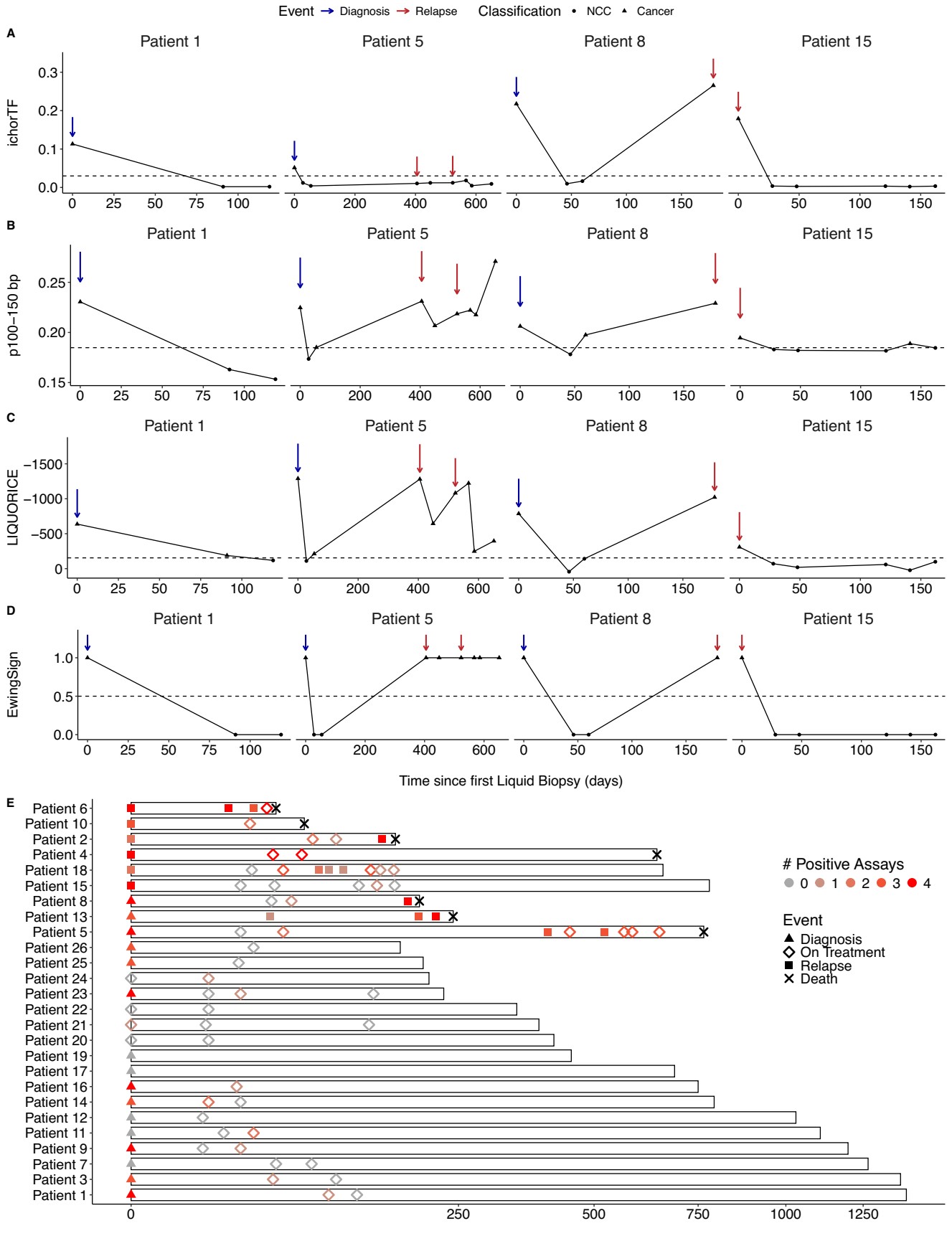

**Figure 5. Longitudinal liquid biopsy monitoring of patients with EwS reflects disease dynamics.**

(A–D) Longitudinal monitoring over time (days) using ichorTF (A), p100–150 bp (B), LIQUORICE (C) and EwingSign (1 = positive for EwS, 0 = negative for EwS, D) for patients 1, 5, 8 and 15. Blue arrows indicate diagnosis events, red arrows indicate relapse events, and an absence of arrows indicates an on-treatment event. Each point indicates a liquid biopsy sample taken, and its result (NCC = circle, Cancer = triangle). (E) Swimmer's plot of all patients in the cohort ($n = 26$). Bar length indicates the length of time the patient has been monitored for, with 0 indicating the date of the first liquid biopsy taken. The shape of the point indicates which event the sample was taken, either diagnosis (triangle), on treatment (diamond) or relapse (square). Death is shown with a black cross at the end of the bar, with the absence of the cross indicating the patient is alive. The colour of the point indicates the number of modalities with a cancer result, ranging from four modalities (red) to no modalities (grey). Values on the x-axis are shown on a square-root scale. Source data are available online for this figure.

each sample were calculated within QSEA using the blind calibration method.

## IchorCNA

cfDNA WGS data were processed with ichorCNA (v0.3.2) to give an estimated tumour fraction (ichorTF) using a panel of normal samples as previously described (Conway et al, 2024). Estimated ichorTF below 0.03 were deemed to be below the limit of detection (Adalsteinsson et al, 2017).

## p100–150 bp

WGS samples were down-sampled to 5 M reads where possible. Methylation enrichment samples were down-sampled to 10 M reads. Counts of fragments with a size between 100 bp and 150 bp were used to calculate the proportion of short fragments.

The threshold of positivity was set equal to the highest p100-150 of the 83 NCCs that were part of cohort 1. The threshold was set separately for WGS and methylation enrichment analyses.

## Quality checks

NGSCheckMate (v1.0.1) (Lee et al, 2017) was used to confirm that all longitudinal samples, as well as enriched and WGS T7-MBD-seq libraries from the same individual, matched as expected. The relative enrichment score (relH) was calculated using the method of the MEDIPs R package (Lienhard et al, 2014) for enriched libraries to confirm enrichment of CpGs, as described by Conway et al. (Conway et al, 2024). The hyperstable fraction (Conway et al, 2024) was also calculated using consistently hypermethylated CpGs (Edgar et al, 2014). Adequate methylation capture was deemed to be achieved when the relH was above 2.5, and the hyperstable fraction was above 0.4. All samples used in this analysis were deemed to have adequate methylation capture.

## DELFI_FTK

Due to the requirement of the employed region-based fragmentomic methods of higher sequencing depth (Cristiano et al, 2019; Peneder et al, 2022, 2021) and increased sensitivity to coverage differences, only the higher-depth sequencing of the 24 WGS cohort 2 NCCs was used, with the cfDNA WGS NCC and cancer samples down-sampled to an average sequencing depth of 1× (0.3–1.2-fold coverage). The down-sampled data served as input for all DELFI_FTK, LOLA and LIQUORICE analyses described below.

Subsequently, to estimate cfDNA fragmentation patterns across the genome, FinaleToolkit (v0.10.7)(Li et al, 2025) was employed to perform a DELFI_FTK analysis, which aims to replicate, in an executable way, the original DELFI method as described by Cristiano et al. (Cristiano et al, 2019). Default parameters were used (Li et al, 2025). Median NCC profile was defined as the median short to long fragment ratio calculated per each 5 Mb genomic bin for the 24 samples in NCC cohort 2.

## LOLA (Locus overlap analysis)

Region-set enrichment analysis was performed as described by Peneder et al (2021) using LOLA (v1.30.0) (Sheffield and Bock, 2016), with the difference of setting bin size to 1 Mb and omitting the CNA filtering step. EwS-specific DNase I hypersensitive sites (DHS) obtained from Peneder et al. (Peneder et al, 2021) were used as an additional region database in the analysis. Significant hits ($q < 0.05$) were summarised across queried samples, and only significant hits appearing in at least eight diagnosis and relapse EwS samples were retained.

## LIQUORICE

The software tool LIQUORICE (v0.5.6)(Peneder et al, 2022, 2021) was employed to profile chromatin accessibility at EWS::FLI1 binding sites (obtained from Peneder et al, 2021) with the default parameters. Additionally, 50-bp read mappability and ichorCNA-based CNA bias correction was used. Universally open DHS sites obtained from Peneder et al. were used as a positive control. The threshold of positivity was set equal to the lower bound of the 95% prediction interval of the 24 tested NCCs from NCC cohort 2, which was calculated with LIQUORICE's summary tool using the default settings (Peneder et al, 2022).

## EwingSign classifier

A methylation-based classifier, EwingSign, was developed using a robust ensemble learning framework adapted from Conway et al (2024) to train a machine learning model capable of predicting three distinct classes: EwS, CIC or non-cancer. The classifier-building steps outlined below were performed within a Nextflow (24.04.2) (Ewels et al, 2020) DSL2 pipeline.

### Methylation array data

We obtained methylation array data for EwS and CIC tissue samples from the Gene Expression Omnibus (Koelsche et al, 2021) (Data Ref: Koelsche et al, 2021), which included ten CIC arrays (eight from the 450k platform and two from the EPIC platform) and 37 EwS arrays (26 from the 450k platform and 11 from the EPIC platform). The SeSaMe R package without pOOBAH (p value

with out-of-band array hybridisation) masking was used to process the IDAT files to obtain beta-values for each probe. These methylation array beta-values were then converted into a window-based read format compatible with T7-MBD-seq, as exemplified by Conway et al (2024). This conversion yielded a QSEA object containing estimated read counts for each sample in windows overlapping the array probes, approximating the counts that would have been obtained if T7-MBD-seq had been performed on the same samples.

### T7-MBD-seq NCC data

We implemented a systematic approach to allocate NCC cfDNA samples across the training, validation, and test sets to ensure the classifier was robust and not biased toward a particular subset of NCCs. Of the 107 total samples, 24 were assigned exclusively to the unseen test set (NCC cohort 2) due to the higher-depth DNA sequencing of these samples, which is required for fragmentomics analysis (Cristiano et al, 2019; Peneder et al, 2022). The remaining 83 samples (NCC cohort 1) were randomly split into training and validation sets at a 75:25 ratio (63 training and 20 validation samples). This process was repeated 25 times, resulting in 25 training–validation NCC splits, which were used together with methylation array data from EwS and CIC tissue samples to train 25 classifiers, each with three classes: EwS, CIC and non-cancer.

### Feature selection

For each of the 25 classifiers, DMRs were identified through pairwise comparisons between EwS tissue samples and the NCC cfDNA training set, CIC tissue samples and the NCC cfDNA training set, and CIC and EwS tissue samples, using the QSEA package. DMRs were selected based on a false discovery rate (FDR) threshold of 0.05. To refine the selection, DMRs where the median beta-value exceeded 0.25 across NCC training set samples were removed. Then, only the DMRs that were common to all three pairwise comparisons were retained. Finally, DMRs were ranked in descending order based on their minimum absolute $\Delta\beta$ (difference in average beta-values) across the three comparisons, and the top 200 were selected as features for classifier training.

### Training 25 ensemble classifiers

To generate the augmented training dataset of in silico mixture samples, fragment counts were mixed either by incorporating an array of EwS or CIC tissue samples (converted to a QSEA object) into an NCC cfDNA training set sample at proportions ranging from 0.005 and 0.1, or by combining two NCC cfDNA training set samples at proportions ranging from 0.15 to 0.5. In total, there were 2200 mixtures of CIC with NCC, 7400 mixtures of EwS with NCC and 10,000 NCC mixtures generated. For each of the 25 classifiers, 100 sub-classifiers were trained using Extreme Gradient Boosting Trees (XGBoost, implemented in the R package xgboost) with default parameters, except for setting the number of trees to 200, sample_size to 0.5 and mtry to 20. Each sub-classifier was trained using the top 200 DMRs (selected as described above) as features and using a subset of the in silico mixture samples, including only those generated from 80% of the array samples and 80% of the NCC samples (selected at random). When applied to a test sample, each sub-classifier results in a prediction score for each of the three classes (these scores sum to one). An ensemble classifier score for each class was then calculated using the mean prediction score

across all 100 sub-classifiers. Then, the EwingSign classifier prediction score for each class was calculated as the mean prediction score across the 25 ensemble classifiers. A sample was assigned as CIC, EwS, or NCC when the EwingSign prediction score for a single class exceeded 0.5. This threshold ensures that the prediction score for the assigned class is greater than the combined prediction score of the other two classes. Samples with all EwingSign class-specific prediction scores below 0.5 were reported as unclassified.

### Evaluation

To assess the sensitivity of the classifier, 28 unseen EwS tissue samples (Patrizi et al, 2024) (Data Ref: Patrizi et al, 2024) were mixed with 24 unseen cohort 2 NCC test set samples. Each mixture sample combination was generated using 26 fixed proportions ranging from 0.001 to 0.05, resulting in a total of 17,472 in silico mixture samples. The EwingSign classifier was then applied to the mixture samples to systematically assess the classifier's sensitivity to low-abundance EwS signals at the 0.5 classification threshold.

Finally, the unseen test set of T7-MBD-seq cfDNA data from 24 NCC samples (NCC cohort 2), 76 EwS samples from 23 patients, and ten CIC samples from three patients was used to independently evaluate the performance of the EwingSign classifier. The performance of EwingSign for cancer detection was assessed agnostically to cancer type by summing the class prediction scores for EwS and CIC together.

---

**The paper explained**

**Problem**

Ewing sarcoma (EwS) is a rare and aggressive cancer with a low tumour mutational burden, making its detection through plasma cell-free DNA (cfDNA) challenging. Current liquid biopsy approaches often lack sensitivity for early diagnosis and relapse monitoring, creating an unmet need for improved non-invasive biomarkers.

**Results**

Eighty-seven plasma cfDNA from 23 patients with EwS and three patients with CIC-rearranged sarcoma were sequenced with T7-MBD seq, a method enabling the recovery of the genome, methylome and fragmentome of cfDNA.

EwingSign, a machine learning model to identify EwS or CIC, was developed, trained and tested using methylome datasets. Applied to plasma cfDNA, EwingSign identified the majority of patients with EwS and CIC at diagnosis and relapse, with no false positive.

cfDNA size distribution, regional and local fragmentation patterns can be modified in cancer patients. Such fragmentomic signals were altered and detectable using T7-MBD seq in most plasma of patients with EwS or CIC at diagnosis and relapse. Integrating methylome and fragmentome data detected all cases at relapse.

**Impact**

Our findings suggest that cfDNA methylome and fragmentome profiling can improve sensitivity for EwS detection and relapse identification. If validated in larger cohorts, this strategy could enable more accurate, minimally invasive disease monitoring and guide timely clinical interventions for patients with EwS.

## Statistics

All statistical tests performed are detailed in the text and figures. Mann–Whitney *U*-tests were performed with Benjamini–Hochberg multiple testing correction using the rstatix R package (v0.7.2, (Kassambara, 2019)). Non-significant *p* values are not plotted. Where detail was not provided, all tests were two-sided with a threshold of *p* < 0.05.

## Graphics

The graphical abstract, Figs. 1 and 3A were created with Biorender.com

# Data availability

Sequencing data are deposited at the European Genome-Phenome Archive (EGA), accession number: EGAD50000002051 (https://ega-archive.org/datasets/EGAD50000002051). A subset of the NCC samples were profiled with T7-MBD-seq in previous studies with sequencing data deposited in EGAS00001005739 and EGAS00001007445. In this study, updated fastq files were used for these samples (due to an update to an internal sample demultiplexing pipeline), and these updated fastqs are included in EGAD50000002051. The Nextflow pipeline for building the EwingSign classifier is accessible from Zenodo (https://doi.org/10.5281/zenodo.17527445). Code to analyse the output of the classifiers and generate all figures is available from Zenodo (https://doi.org/10.5281/zenodo.17527449).

The source data of this paper are collected in the following database record: biostudies:S-SCDT-10_1038-S44321-026-00396-7.

# Peer review information

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

## Acknowledgements

This work was funded by the CRUK National Biomarker Centre (CTRNBC-2022/100001), The Christie Charitable Trust and the Children's Cancer Research Fund (DIVE/2023). Support was received from the Manchester Experimental Cancer Medicines Centre and the National Institute for Health and Care Research (NIHR) Manchester Biomedical Research Centre (BRC) (NIHR203308). The authors thank Ekram Aidaros-Talbot, Lee-Anne van Winkel, Matthew Lancashire, and all the members of the NBC for their constant help and support. The authors thank the Manchester Cancer Research Centre Biobank and Royal Manchester Children's Hospital for assistance with sample collection and processing, and Holly Cassell for training in bioinformatic pipelines. The authors thank the Core Facilities at the Cancer Research UK Manchester Institute (C5759/A27412), with particular thanks to the Molecular Biology Core Facility and the Scientific Computing Team. The authors also would like to thank the healthy volunteers, patients with EwS and CIC, and their families for consenting to this study.

## Author contributions

**Sophie A Richardson**: Conceptualisation; Data curation; Software; Formal analysis; Validation; Investigation; Visualisation; Methodology; Writing—original draft; Project administration; Writing—review and editing. **Aram Safrastyan**: Data curation; Software; Formal analysis; Validation; Investigation; Visualisation; Methodology; Writing—original draft; Writing—review and editing. **Mina Karimpour**: Data curation; Software; Formal analysis; Validation; Investigation; Visualisation; Methodology; Writing—original draft; Writing—review and editing. **Gayatri Gulati**: Data curation; Investigation; Writing—review and editing. **Patrick JB Harker**: Data curation; Software; Writing—review and editing. **Alan Redfern**: Data curation; Project administration; Writing—review and editing. **Simon P Pearce**: Software; Writing—review and editing. **Vsevolod J Makeev**: Resources; Writing—review and editing. **Bernadette Brennan**: Resources; Writing—review and editing. **Alexander TJ Lee**: Resources; Writing—review and editing. **Alexandra Clipson**: Data curation;

Writing—review and editing. **Steven M Hill**: Data curation; Software; Formal analysis; Supervision; Validation; Writing—review and editing. **Caroline Dive**: Supervision; Funding acquisition; Writing—review and editing. **Dominic G Rothwell**: Supervision; Funding acquisition; Writing—review and editing. **Martin G McCabe**: Conceptualisation; Resources; Supervision; Funding acquisition; Project administration; Writing—review and editing. **Florent Mouliere**: Conceptualisation; Resources; Formal analysis; Supervision; Funding acquisition; Investigation; Visualisation; Writing—original draft; Writing—review and editing.

Source data underlying figure panels in this paper may have individual authorship assigned. Where available, figure panel/source data authorship is listed in the following database record: biostudies:S-SCDT-10_1038-S44321-026-00396-7.

## Disclosure and competing interests statement

SMH, SPP, AC, CD, DGR and FM are co-inventors on patents related to cell-free DNA analysis. FM has consulted for Roche Dx. AC and FM have received support from Biomodal. CD has received research funding/educational research grants from Amgen, AstraZeneca, Astex Pharmaceuticals, Angle PLC, Boehringer Ingelheim, Carrick Therapeutics, Celgene, Clearbridge Biomedics, CV6 Therapeutics (NI) Ltd, Epigene Therapeutics Inc, GSK, Guardant, Menarini, Merck AG, Neomed Therapeutics, Novartis, RedX Pharma, Roche, Taiho Oncology, Thermo Fisher Scientific and UCB Pharma. CD has received honoraria for consultancy and/or advisory boards from Biocartis, Merck, AstraZeneca, GRAIL and Boehringer Ingelheim. ATJL has received honoraria and support with conference attendance from Alexion Pharmaceuticals. The remaining authors declare no competing interests.

