## [Peer Review File · EMBO Molecular Medicine]

Cell-free DNA methylome and fragmentome analysis for relapse monitoring of Ewing Sarcoma

Sophie Richardson, Aram Safrastyan, Mina Karimpour, Gayatri Gulati, Patrick Harker, Alan Redfern, Simon Pearce, Vsevolod Makeev, Bernadette Brennan, Alexander Lee, Alexandra Clipson, Steven Hill, Caroline Dive, Dominic Rothwell, Martin McCabe, and Florent Mouliere

Corresponding authors: Florent Mouliere (florent.mouliere@cruk.manchester.ac.uk) , Martin McCabe (Martin.McCabe@manchester.ac.uk)

Review Timeline:

Submission Date:	23rd Jun 25
Editorial Decision:	18th Jul 25
Revision Received:	25th Nov 25
Editorial Decision:	12th Jan 26
Revision Received:	20th Jan 26
Accepted:	29th Jan 26

Editor: Lise Roth

Transaction Report:

18th Jul 2025

Dear Florent,

Thank you for submitting your study to EMBO Molecular Medicine. We have now received feedback from the three reviewers who agreed to evaluate your manuscript. As you will see from the reports below, the referees acknowledge the interest of the study and are overall supporting publication of your work pending appropriate revisions.

We further consulted with the referees regarding referee #3's request to add more patients' samples and agreed that this would NOT be requested for further consideration. We would nevertheless strongly encourage you to include additional age-matched healthy samples.

Addressing the other reviewers' concerns in full will be necessary for further considering the manuscript in our journal, and acceptance of the manuscript will entail a second round of review. EMBO Molecular Medicine encourages a single round of revision only and therefore, acceptance or rejection of the manuscript will depend on the completeness of your responses included in the next, final version of the manuscript. For this reason, and to save you from any frustrations in the end, I would strongly advise against returning an incomplete revision.

We are expecting your revised manuscript within three months, if you anticipate any delay, please contact us.

We require:

- 1) A .docx formatted version of the manuscript text (including legends for main figures, EV figures and tables). Please make sure that the changes are highlighted to be clearly visible.
- 2) Individual production quality figure files as .eps, .tif, .jpg (one file per figure). For guidance, download the 'Figure Guide PDF' (<https://www.embopress.org/page/journal/17574684/authorguide#figureformat>).
- 3) At EMBO Press we ask authors to provide source data for the main manuscript figures. You will receive a separate email with instructions for providing source data with your revised manuscript, including how to upload and organize the files.

Additional information on source data and instruction on how to label the files are available

- 4) A .docx formatted letter INCLUDING the reviewers' reports and your detailed point-by-point responses to their comments. As part of the EMBO Press transparent editorial process, the point-by-point response is part of the Review Process File (RPF), which will be published alongside your paper.

- 5) A complete author checklist, which you can download from our author guidelines (<https://www.embopress.org/page/journal/17574684/authorguide#submissionofrevisions>). Please insert information in the checklist that is also reflected in the manuscript. The completed author checklist will also be part of the RPF.

- 6) All Materials and Methods need to be described in the main text using our 'Structured Methods' format. According to this format, the Methods section includes a Reagents and Tools Table (listing key reagents, experimental models, software and relevant equipment and including their sources and relevant identifiers) followed by a Methods and Protocols section describing the methods, ideally using a step-by-step protocol format. The aim is to facilitate adoption of the methodologies across labs. Please download and fill our Reagents and Tools Table template (.docx), which you can find in our author guidelines: <https://www.embopress.org/page/journal/14693178/authorguide#structuredmethods>.

- 7) Please note that all corresponding authors are required to supply an ORCID ID for their name upon submission of a revised manuscript.

- 8) It is mandatory to include a 'Data Availability' section after the Materials and Methods. Before submitting your revision, primary datasets produced in this study need to be deposited in an appropriate public database, and the accession numbers and database listed under 'Data Availability'. Please remember to provide a reviewer password if the datasets are not yet public (see

<https://www.embopress.org/page/journal/17574684/authorguide#dataavailability>).

9) For data quantification: please specify the name of the statistical test used to generate error bars and P values, the number (n) of independent experiments (specify technical or biological replicates) underlying each data point and the test used to calculate p-values in each figure legend. The figure legends should contain a basic description of n, P and the test applied. Graphs must include a description of the bars and the error bars (s.d., s.e.m.). Please provide exact p values.

10) Our journal encourages inclusion of *data citations in the reference list* to directly cite datasets that were re-used and obtained from public databases. Data citations in the article text are distinct from normal bibliographical citations and should directly link to the database records from which the data can be accessed. In the main text, data citations are formatted as follows: "Data ref: Smith et al, 2001" or "Data ref: NCBI Sequence Read Archive PRJNA342805, 2017". In the Reference list, data citations must be labeled with "[DATASET]". A data reference must provide the database name, accession number/identifiers and a resolvable link to the landing page from which the data can be accessed at the end of the reference. Further instructions are available at .

11) We replaced Supplementary Information with Expanded View (EV) Figures and Tables that are collapsible/expandable online. EV Figures should be cited as 'Figure EV1, Figure EV2' etc... in the text and their respective legends should be included in the main text after the legends of regular figures.

12) The paper explained: EMBO Molecular Medicine articles are accompanied by a summary of the articles to emphasize the major findings in the paper and their medical implications for the non-specialist reader. Please provide a draft summary of your article highlighting

13) Author contributions: CRediT has replaced the traditional author contributions section because it offers a systematic machine readable author contributions format that allows for more effective research assessment. Please remove the Authors Contributions from the manuscript and use the free text boxes beneath each contributing author's name in our system to add specific details on the author's contribution. More information is available in our guide to authors.

Please also suggest a visual abstract to illustrate your article as a PNG file 550 px wide x 300-600 px high. A cropped portion of this image will serve as thumbnail for the table of content on our webpage.

16) As part of the EMBO Publications transparent editorial process initiative (see our Editorial at <http://embomolmed.embopress.org/content/2/9/329>), EMBO Molecular Medicine will publish online a Review Process File (RPF) to accompany accepted manuscripts.

In the event of acceptance, this file will be published in conjunction with your paper and will include the anonymous referee reports, your point-by-point response and all pertinent correspondence relating to the manuscript. Let us know whether you agree with the publication of the RPF and as here, if you want to remove or not any figures from it prior to publication.

I look forward to receiving your revised manuscript.

With kind regards,

Lise

***** Reviewer's comments *****

Referee #1 (Comments on Novelty/Model System for Author):

The work is of high technical quality. The authors use appropriate cfDNA extraction tools, library preparation methods and sequencing instruments. The novelty is high as they applied cfDNA analysis in an under-studied cancer type. Given that this is a pilot study, the work is underpowered to measure true sensitivity, specificity, etc. In addition, how the 4 features would be integrated together to help guide clinicians in improving patient care remains to be determined. Thus, it is difficult to assess Medical impact. The model system is adequate.

Referee #1 (Remarks for Author):

The authors performed a pilot study exploring many cancer-associated features of cfDNA (fragmentation, methylation, copy number and coverage) in an under-studied cancer type (EwS). The work is important, and the methods seem appropriate. However, many aspects of the work need to be worked on prior to publication.

Comments:

Throughout the manuscript, ensure that abbreviations are spelled out first. e.g. T7-MBD seq, CIC-rearranged (what is CIC?) in the abstract.

Abstract: why is EwS detection difficult in plasma? The authors mention the difficulty and do not cite any literature statistics in the introduction.

Introduction: The authors should describe in more detail the T7-MBD seq assay. Is it probe enrichment? Antibody? The authors should specify that the cfDNA is split for WGS and methylation assays.

F1: I'm struggling to follow the figure numbers with the text numbers. There are a total of 107 NCCs in this study, but we only see the 29 unseen test cohort in F1.

The authors mention that 98 samples were previously sequencing in another study and 9 are new. How are the old/new samples distributed in the seen and unseen test cohorts? The authors would need to look for batch effects among the different sets of NCCs

Section 2.2: Define LOLA.

"p100-150 bp was deemed to be positive if the value exceeded the threshold of 0.209." Why this threshold? What statistics support this cutoff?

F2A: Add chromosome numbers to help readers

F2A: legend can be refined. The standard copy number of autosomal chromosomes in humans is 2. Do you mean Copy number variation? If so, did no samples have copy number variants greater than 1? It's also difficult to follow the y-axis and when the data changes from Patient 1 to Patient 2, etc. Which patients are CIC? Which are EWS?

F2C,D,F: Indicate which are CIC and which are EWS.

F2C: indicate how many samples are used to obtain the traces.

Why are only 5 NCC samples processed through LIQUORICE? What happened to the other 24?

The sentences: "Liquid biopsies are bio-fluids collected from the body, most commonly blood, and are minimally invasive (Janssen et al, 2024). They contain cell-free DNA (cfDNA), RNA and other analytes released into the bloodstream (Song et al, 2022)." reads very awkwardly for me. Do the liquid biopsies contain cfDNA, RNA and other analytes? No, the bio-fluid contains these markers, and the authors are performing a cfDNA-based liquid biopsy. I recommend adjusting the language.

"Cancer cells also release DNA into the bloodstream, referred to as circulating tumour DNA (ctDNA) (van der Pol & Mouliere, 2019)." I agree that van der Pol & Mouliere discuss this in their review, but this observation was first made by Mandel & Metais in 1948.

"Using a DELFI-like approach, the ratio of short to long fragments across the genome in 5 Mb bins suggested an enrichment in short fragments across chromosome 8q in diagnosis and relapse samples compared to first-line treatment (Fig EV3a)." Is missing a citation for DELFI, and the authors should consider the broad readership of EMBO Mol. Med. and consider explaining what "DELFI" is.

"Unsupervised clustering of DELFI analysis shows separation of 8/12 diagnosis and 12/18 relapse events from NCCs (Fig EV3b)."

Is the analysis "DELFI", as described in the original Nature paper, or is it "DELFI-like", where the authors made changes to the method? This should be much better explained in the manuscript as it is not clear what the authors did in this portion of the paper.

How do the EwigSign scores correlate with tumor fractions? If the classifier is built on tumor-specific DMRs, I would expect some correlation. I see this was done in Fig. EV4 but the authors should also compare to their ichorCNA TFs.

Please make the manuscript easier to review. Figure captions in line with the actual figure, a single PDF with all the supplementary figures, etc.

Referee #2 (Comments on Novelty/Model System for Author):

The technical quality of the study is high, with well-executed methodologies, appropriate use of statistical tools, and comprehensive multimodal cfDNA profiling. The novelty is also high, as the authors systematically compare multiple non-mutation-based liquid biopsy strategies-methylation, fragmentomics, and chromatin accessibility-for minimal residual disease detection in a tumour type with low genomic complexity, which represents a significant conceptual advance. The medical impact is moderate at this stage, as the study is exploratory and retrospective, with a relatively small and heterogeneous cohort, but the findings are promising and could inform future prospective trials. The model system, based on patient-derived plasma cfDNA from individuals with Ewing Sarcoma and CIC-rearranged sarcomas, is appropriate. However, the use of non-age-matched controls and the limited cohort size constrain generalisability.

Referee #2 (Remarks for Author):

This manuscript presents a scientifically compelling comparative study of cfDNA-based methodologies for detecting tumour-derived signal in Ewing Sarcoma (EwS) and CIC-rearranged sarcomas. Instead of relying solely on fusion or mutation-based assays, which may have limited sensitivity in paediatric sarcomas with low mutational burden, the authors explore a tumour-agnostic approach centred on epigenomic and fragmentomic features. Specifically, they evaluate copy number alterations (ichorCNA), global and locus-specific cfDNA fragmentation (p100-150 and LIQUORICE), and cfDNA methylation patterns using T7-MBD-seq, including a novel classifier (EwigSign) trained on tumour-derived methylation profiles.

This work is of particular interest because it systematically compares the performance of these orthogonal cfDNA features for detecting disease at diagnosis and relapse. The study provides valuable insight into the utility of non-mutation-based ctDNA detection strategies for broader clinical implementation. The manuscript is clearly written, methodologically detailed, and addresses a pressing clinical need in the field of paediatric oncology. However, there are several major points that warrant further clarification or revision prior to acceptance:

Major Comments

1. Non-age-matched control population

The non-cancer control (NCC) cohort has a median age of 62 years versus 22 years in the patient cohort. As cfDNA fragmentation and methylation are known to vary with age, this mismatch could artificially increase classifier performance. The authors should: 1) Expand their discussion of this limitation and 2) Evaluate whether age correlates with fragment length or classifier score among controls.

2. Lack of benchmarking against molecular gold standards

The authors correctly note that prior studies have used detection of the EWSR1::FLI1 fusion for disease monitoring. However, this manuscript does not compare their approach to those gold standards, either in terms of sensitivity or lead time to detection. The authors should clarify: 1) Whether ddPCR or targeted fusion detection was available for any patients in this cohort, 2) How the sensitivity of EwingSign compares to those established methods in similar clinical contexts, 3) If not available, this should be clearly acknowledged as a limitation.

3. Concordance between modalities

The authors employ four orthogonal cfDNA-derived biomarkers: ichorTF (CNA), p100-150 (fragmentomics), LIQUORICE (TF footprinting), and EwingSign (methylation). However, no formal comparison of overlap or complementarity is provided. This is particularly relevant given that LIQUORICE identified two relapse samples missed by all other methods. 1) The authors should include overlap statistics (e.g., Venn diagrams, Cohen's κ) across modalities and, 2) A brief analysis of whether discordant samples share biological or technical features (e.g., low ctDNA, tumour subtype, sampling time) would be valuable.

4. Threshold derivation and ROC presentation

Although thresholds for positivity are defined using ROC analyses, these are relegated to Supplementary Figure EV6. Given the central role these thresholds play in classifier interpretation and subsequent performance comparisons, it would be more appropriate to present the ROC curves in the main figures. This would improve transparency and help readers assess the diagnostic value of each modality.

5. Selection criteria for illustrative patients

The four exemplar patients in Figure 4A-D are useful, but the rationale for their selection should be stated. Do they reflect typical, best-case, or edge-case scenarios?

6. Lead-time calculation. The authors state that the lead time between molecular and clinical relapse could not be estimated due to the retrospective design and inconsistent sampling. However, Figure 4A-D clearly shows representative cases with positive cfDNA signals prior to radiological relapse. Even if a cohort-wide estimation is not feasible, the authors should report individual lead times (in days or weeks) for these patients and comment on the potential clinical implications. A median and range for those with available data would provide valuable insight into the potential of cfDNA assays for anticipatory intervention.

Minor comments:

1. Incomplete Visualisation in Supplementary Figure EV5

Figure EV5 is a key supplementary figure supporting longitudinal analyses. However, in the axis labels, numerical tick marks are overlapping, this should be improved. since limits interpretability.

Referee #3 (Comments on Novelty/Model System for Author):

Technical quality: multiple testing correction is missing, authors should report on cfDNA quality control (fragment analysis).

Novelty: multivariate analysis of cfDNA is novel, however in the current manuscript actual integration of the different data layers is not performed.

Medical impact: this is a proof-of-concept study with a limited number of cases/samples and therefore not clear yet how impactful this approach will be on patients.

Referee #3 (Remarks for Author):

In this manuscript the authors present a proof-of-concept study to use cfDNA methylomics and fragmentomics for relapse monitoring in pediatric sarcoma, including Ewing Sarcoma and CIC-rearranged sarcoma. Both omics layers are generated using T7-MBD-seq and analysed using ichorCNA (for ichorTF), LOLA for size distribution analysis of cfDNA, LIQUORICE at the EWS-FLI1 binding regions as well as methylation based classification. These different blood-based analytes show promise for monitoring patients with EwS or CIC and detection of relapse events, however larger patient cohorts are needed to prove clinical utility and to allow better evaluation of the multi-analyte approach for disease monitoring, which is my biggest concern and comment on the current study.

1. Overview Figure 1 should also mention the 78 other NCCs (non-cancer controls) that are used for training, and also the in

- silico mixtures for validation (from Patrizi et al, 2024). In addition, the results from the latter validation cohort (in Fig EV4A) are important to my opinion and therefore I would advise to include this figure in the main manuscript.
2. In paragraph 2.2, different analytes (eg ichorTF, LIQUORICE SCORE, etc) are tested in different comparisons, requiring multiple testing correction which seems not to be done.
 3. At least a few (almost) age-matched NCCs should be included in this study to validate the real performance of the classifier/analytes.
 4. On the DELFI analysis: why is chromosome 8q mentioned here? What is the rationale to pick this one? What about other regions?
 5. Both for the LIQUORICE and the EwingSign score, it is not clear how the cut-off of -105.6 and 50%, respectively, is determined. Was this done in a completely unbiased way? Authors should elaborate more on this.
 6. While interesting proof-of-concept results are obtained in this study, the multimodal approach, integrating the 4 different analytes is missing in this study. The performance of the different analytes is described separately, however the manuscript would benefit from a more integrated analysis (both in 2.2 and 2.3). As it will probably be difficult to properly perform this with the current small sample size, authors should consider to increase sample numbers, eg by involving other centers.
 7. One of the last conclusions from the results is that "at least one modality was positive in 17/18 relapse samples": however it was not mentioned what was the result for non-relapse cases and for NCCs? This should also be described.
 8. The performance of the p100-150 measures seems to be a bit more doubtful, with false positive detection and the least performance in the different study parts. Discussion on this is missing in the manuscript (cfr Sup Table 7).
 9. Figure 4: the crosses indicating whether a patient died or not, are missing in the figure
 10. QC (Tapestation) of the cfDNAs should be reported. Where only good quality samples included or also samples with a lot of high molecular weight DNA?

***** Reviewer's comments *****

Referee #1 (Comments on Novelty/Model System for Author):

The work is of high technical quality. The authors use appropriate cfDNA extraction tools, library preparation methods and sequencing instruments. The novelty is high as they applied cfDNA analysis in an under-studied cancer type. Given that this is a pilot study, the work is underpowered to measure true sensitivity, specificity, etc. In addition, how the 4 features would be integrated together to help guide clinicians in improving patient care remains to be determined. Thus, it is difficult to assess Medical impact. The model system is adequate.

Referee #1 (Remarks for Author):

The authors performed a pilot study exploring many cancer-associated features of cfDNA (fragmentation, methylation, copy number and coverage) in an under-studied cancer type (EwS). The work is important, and the methods seem appropriate. However, many aspects of the work need to be worked on prior to publication.

We thank the Reviewer for their time and constructive comments. We have detailed replies to their comments below.

Comments:

Throughout the manuscript, ensure that abbreviations are spelled out first. e.g. T7-MBD seq, CIC-rearranged (what is CIC?) in the abstract.

We have detailed the abbreviations through the manuscript, including in the abstract.

Abstract: why is EwS detection difficult in plasma? The authors mention the difficulty and do not cite any literature statistics in the introduction.

Due to the space constraint of the abstract, this is difficult to detail. However, we have detailed in the introduction some of the challenges related to the detection of cancer in patients with EwS with liquid biopsy.

The modified sentence is on line 64 and reads as follow: "Due to its low mutational burden (Crompton et al, 2014; Brohl et al, 2014), detection of genetic mutations in EwS can be challenging. "

Introduction: The authors should describe in more detail the T7-MBD seq assay. Is it probe enrichment? Antibody? The authors should specify that the cfDNA is split for WGS and methylation assays.

The method T7-MBD seq has been further detailed in the introduction. The corresponding sentence reads as follow: "...T7-MBD-seq assay, an enrichment-based methylation capture method using a methyl binding domain protein (Conway et al, 2024; Chemi et al, 2022). T7-MBD-seq is an integrated workflow combining genome wide methylation analysis and (shallow) whole genome sequencing (WGS) for the recovery of CNAs and cfDNA fragmentomic features."

The T7-MBD seq approach is described in the Methods section in detail. The split of cfDNA between WGS and methylation capture is detailed in the Methods section 4.4. The corresponding sentence reads as follow: "90% of the sample pool underwent a methylation enrichment with the EpiMark® Methylated DNA Enrichment Kit (New

England Biolabs) with the remaining 10% of the library pool retained for WGS". This has also been illustrated in the revised Figure 1.

F1: I'm struggling to follow the figure numbers with the text numbers. There are a total of 107 NCCs in this study, but we only see the 29 unseen test cohort in F1.

We apologize if this was unclear. We included 17 additional NCCs in the revised manuscript and removed 17 NCCs that were aged ≥ 70 years, and we now have a total of 107 NCCs. Of the 107 NCCs samples, 83 were included in the training / validation cohort, and 24 were held out in an unseen test set. This has been amended in the revised Figure 1.

The authors mention that 98 samples were previously sequencing in another study and 9 are new. How are the old/new samples distributed in the seen and unseen test cohorts? The authors would need to look for batch effects among the different sets of NCCs

We have now included the training NCCs in Fig 1 for clarity. The source of the NCCs and whether they belong to the test or training set can be found in TableEV3.

Section 2.2: Define LOLA.

LOLA (Locus Overlap Analysis) has now been defined section 4.11 of the Methods.

"p100-150 bp was deemed to be positive if the value exceeded the threshold of 0.209." Why this threshold? What statistics support this cutoff?

This cut-off was derived from ROC analysis, selecting a cut-off value that gives $\geq 90\%$ specificity. To make this clearer, ROC curves have been moved to the main figures (Fig 4) from FigEV6. In the current revised version of the manuscript, the threshold was calculated based on the highest p100-150 achieved by the cohort 1 83 NCCs not part of the main analysis and the corresponding section currently reads as follows: "The threshold of positivity was set equal to the highest p100-150 of the 83 NCCs that were part of cohort 1 The threshold was set separately for WGS and methylation enrichment analyses."

F2A: Add chromosome numbers to help readers

This has been included in the revised Figure 2A.

F2A: legend can be refined. The standard copy number of autosomal chromosomes in humans is 2. Do you mean Copy number variation? If so, did no samples have copy number variants greater than 1? It's also difficult to follow the y-axis and when the data changes from Patient 1 to Patient 2, etc. Which patients are CIC? Which are EWS?

The Figure 2A and the corresponding legend has been modified to indicate the scale is from -1 to 1 on a log scale. An annotation indicating the type of pathology was added to the plot. The updated text legend for Figure 2A reads as follow: "A: Heatmap of CNAs. Each row is a sample with patient and pathology annotated on the right. Each column is a region of the genome (1,000,000 bp windows) and separate chromosomes denoted by the top annotation. Gains = red. Losses = blue."

F2C,D,F: Indicate which are CIC and which are EWS.

Figure 2C and E have been updated to indicate the EwS and CIC results. We have included a new Supplementary Figure EV4 which is detailing the results for CIC and EwS from Figure 2B, C and F.

F2C: indicate how many samples are used to obtain the traces.

The sample number has been added to the Figure 2C and corresponding Figure legend.

Why are only 5 NCC samples processed through LIQUORICE? What happened to the other 24?

In the previous version of our manuscript, the majority of our NCCs were sequenced to low coverage (~0.1X). LIQUORICE, LOLA and DELFI_FTK analysis all require higher coverage (> 1.5X based on internal benchmarking). Therefore, we used only the 5 NCCs where higher coverage WGS was available for this analysis. This has been modified in the revised manuscript and sequenced deeper the 24 NCCs included in the LIQUORICE analysis.

The sentences: "Liquid biopsies are bio-fluids collected from the body, most commonly blood, and are minimally invasive (Janssen et al, 2024). They contain cell-free DNA (cfDNA), RNA and other analytes released into the bloodstream (Song et al, 2022)." reads very awkwardly for me. Do the liquid biopsies contain cfDNA, RNA and other analytes? No, the bio-fluid contains these markers, and the authors are performing a cfDNA-based liquid biopsy. I recommend adjusting the language.

This has been rephrased and reads as follow in the revised manuscript: "Liquid biopsies are bio-fluids collected from the body, most commonly blood, and are minimally invasive (Janssen et al, 2024). These bio-fluids contain cell-free DNA (cfDNA), cell-free RNA and other molecules that can be analysed with molecular and biochemical techniques (Song et al, 2022)"

"Cancer cells also release DNA into the bloodstream, referred to as circulating tumour DNA (ctDNA) (van der Pol & Mouliere, 2019)." I agree that van der Pol & Mouliere discuss this in their review, but this observation was first made by Mandel & Metais in 1948.

We thank the reviewer for this comment. The pioneer work of Mandel & Metais do not discuss of DNA but more broadly of nucleic acids. We have cited the work from Leon et al, 1977 (PMID837366), which is the first publication, to our knowledge, to identify cfDNA in the cancer context.

"Using a DELFI-like approach, the ratio of short to long fragments across the genome in 5 Mb bins suggested an enrichment in short fragments across chromosome 8q in diagnosis and relapse samples compared to first-line treatment (Fig EV3a)." Is missing a citation for DELFI, and the authors should consider the broad readership of EMBO Mol. Med. and consider explaining what "DELFI" is.

We modified this paragraph which now reads as follow: "Region-based cfDNA fragmentation analysis was also performed to gain a deeper understanding of the observed global fragmentation shifts. To mitigate potential sequencing coverage based confounding effects, coverage was normalised by down sampling both NCC and cancer WGS samples to ~1x. Subsequently, an approach similar to DELFI (Cristiano et al, 2019) was used to assess the ratio of short to long fragments across

the genome in 5 Mb bins (DELFI_FTK) (Li et al, 2024). This analysis suggested an enrichment in short fragments across chromosome arms 8p and 8q, which commonly have copy number gains in EwS (Tirode et al, 2014; Fig 2A), in diagnosis and relapse samples compared to first-line treatment (Fig EV6A). Additionally, unsupervised clustering of DELFI_FTK analysis showed separation of 9/16 diagnosis and 12/18 relapse events from NCCs (Fig EV6B)."

"Unsupervised clustering of DELFI analysis shows separation of 8/12 diagnosis and 12/18 relapse events from NCCs (Fig EV3b)."

Is the analysis "DELFI", as described in the original Nature paper, or is it "DELFI-like", where the authors made changes to the method? This should be much better explained in the manuscript as it is not clear what the authors did in this portion of the paper.

DELFI, as detailed in the original Cristiano et al article, is a closed technology with non-executable code. We have therefore used a "DELFI-like" approach available as part of the finaletoolkit repository (<https://github.com/epifluidlab/FinaleToolkit>). To avoid confusion, we have changed the name to "DELFI_FTK" and further explained this section of the manuscript. We have included the following sentence: "Subsequently, an approach similar to DELFI (Cristiano et al, 2019), was used to assess the ratio of short to long fragments across the genome in 5 Mb bins (DELFI_FTK) (Li et al, 2024)."

We also included in the Methods section details on how this analysis was done. The corresponding section reads as follow: "Subsequently, to estimate cfDNA fragmentation patterns across the genome, FinaleToolkit (v0.10.7)(Li et al, 2024) was employed to perform a DELFI_FTK analysis, which aims to replicate the original DELFI (Cristiano et al, 2019)."

How do the EwigSign scores correlate with tumor fractions? If the classifier is built on tumor-specific DMRs, I would expect some correlation. I see this was done in Fig. EV4 but the authors should also compare to their ichorCNA TFs.

A direct correlation of the EwigSign prediction score and ichorTF is not appropriate to perform, as ichorTF is a quantitative metric and EwigSign a mean probability. We can however compare the 2 metrics (see Figure below for reviewers only). We can observe that the EwigSign prediction is high when ichorTF is higher than 3%. But also, that 19 samples are correctly detected by EwigSign when their ichorTF is below the limit of detection.

Please make the manuscript easier to review. Figure captions in line with the actual figure, a single PDF with all the supplementary figures, etc.

We thank the reviewer for this comment. The figure captions and figures were submitted as per the guidelines for EMBO Molecular Medicine.

Referee #2 (Comments on Novelty/Model System for Author):

The technical quality of the study is high, with well-executed methodologies, appropriate use of statistical tools, and comprehensive multimodal cfDNA profiling. The novelty is also high, as the authors systematically compare multiple non-mutation-based liquid biopsy strategies-methylation, fragmentomics, and chromatin accessibility-for minimal residual disease detection in a tumour type with low genomic complexity, which represents a significant conceptual advance. The medical impact is moderate at this stage, as the study is exploratory and retrospective, with a relatively small and heterogeneous cohort, but the findings are promising and could inform future prospective trials. The model system, based on patient-derived plasma cfDNA from individuals with Ewing Sarcoma and CIC-rearranged sarcomas, is appropriate. However, the use of non-age-matched controls and the limited cohort size constrain generalisability.

Referee #2 (Remarks for Author):

This manuscript presents a scientifically compelling comparative study of cfDNA-based methodologies for detecting tumour-derived signal in Ewing Sarcoma (EwS) and CIC-rearranged sarcomas. Instead of relying solely on fusion or mutation-based assays, which may have limited sensitivity in paediatric sarcomas with low mutational burden, the authors explore a tumour-agnostic approach centred on

epigenomic and fragmentomic features. Specifically, they evaluate copy number alterations (ichorCNA), global and locus-specific cfDNA fragmentation (p100-150 and LIQUORICE), and cfDNA methylation patterns using T7-MBD-seq, including a novel classifier (EwingSign) trained on tumour-derived methylation profiles.

This work is of particular interest because it systematically compares the performance of these orthogonal cfDNA features for detecting disease at diagnosis and relapse. The study provides valuable insight into the utility of non-mutation-based ctDNA detection strategies for broader clinical implementation. The manuscript is clearly written, methodologically detailed, and addresses a pressing clinical need in the field of paediatric oncology. However, there are several major points that warrant further clarification or revision prior to acceptance:

We thank the Reviewer for their time and constructive comments. We have detailed replies to their comments below.

Major Comments

1. Non-age-matched control population

The non-cancer control (NCC) cohort has a median age of 62 years versus 22 years in the patient cohort. As cfDNA fragmentation and methylation are known to vary with age, this mismatch could artificially increase classifier performance. The authors should: 1) Expand their discussion of this limitation and 2) Evaluate whether age correlates with fragment length or classifier score among controls.

We agree with the reviewer that it is an important point. In the revised manuscript, we included 17 additional NCCs and removed 17 NCCs that were aged ≥ 70 years, and we have now in total 107 NCCs. 20/107 were below 45 years old, and therefore below the 50 years old cut-off previously reported in the landmark Peneder et al (2021) publication.

- 1. The following sentence to discuss this limitation is present the discussion: "Additionally, our NCC cohort (median age = XX years) was not age- or risk-matched, and is significantly older ($p < 0.01$, Wilcoxon test) than our patient cohort (median age = 22 years) due to the challenge of obtaining NCCs from a younger population. "*
- 2. We investigated the impact of the age on the fragment length (cf Figure for review below). We observe that the proportion of cfDNA fragments P100-150bp is increasing depending on the age-range of the NCCs. However, the difference with samples collected from patients with EwS at relapse remain significant irrespective of the age range.*

2. Lack of benchmarking against molecular gold standards

The authors correctly note that prior studies have used detection of the EWSR1::FLI1 fusion for disease monitoring. However, this manuscript does not compare their approach to those gold standards, either in terms of sensitivity or lead time to detection. The authors should clarify: 1) Whether ddPCR or targeted fusion detection was available for any patients in this cohort, 2) How the sensitivity of EwingSign compares to those established methods in similar clinical contexts, 3) If not available, this should be clearly acknowledged as a limitation.

It is difficult to consider ddPCR fusion assays as gold standard, due to the variability in EWSR1 fusion structure and the necessity to design custom build fusion assays for detecting such signal in plasma samples. We feel this is outside the scope of this specific manuscript. However, we agree it would be interesting to compare our results to such assays, and we have included the following sentence in the discussion: "Therefore, we built upon the research of Peneder et al. (Peneder et al, 2021) to determine the utility of cfDNA fragmentation as a tool for monitoring EwS patients, although a limitation of our study is a lack of comparison between our methods and those for detecting molecular alterations or EwS fusion molecules."

3. Concordance between modalities

The authors employ four orthogonal cfDNA-derived biomarkers: ichorTF (CNA), p100-150 (fragmentomics), LIQUORICE (TF footprinting), and EwingSign (methylation). However, no formal comparison of overlap or complementarity is provided. This is particularly relevant given that LIQUORICE identified two relapse samples missed by all other methods. 1) The authors should include overlap statistics (e.g., Venn diagrams, Cohen's κ) across modalities and, 2) A brief analysis of whether discordant samples share biological or technical features (e.g., low ctDNA, tumour subtype, sampling time) would be valuable.

We thank the reviewer for this comment.

1. We have included a heatmap indicating which samples were positive for each of the modalities tested in Figure 4B. This Figure can be visualised in a similar manner to a Venn diagram, but allows for the discrimination of individual samples, which timepoint they are from, and whether the patient has CIC or EwS (correlation between all modalities).
2. There are both biological and technical explanations to the difference between the modalities. All samples with a high tumour fraction (>3% by ichorCNA) are detected by EwingSign, but 19 samples with a low tumour fraction are only detected by EwingSign. LIQUORICE is missing more patients with CIC, also in cases with high tumour fraction by ichorCNA (1 patient at baseline, 2 at relapse). This could be explained by the database used to tailor the LIQUORICE analysis which was more focused toward EwS. The concordance is high at baseline between EwingSign and P100-150bp, and then is decreasing at relapse, with some cases solely detected with one of the modalities. This could reflect the partially different biological mechanisms reflected by either EwingSign (the tissue of origin of cfDNA) or P100-150bp (reflecting both the origin of cfDNA as well as cell-death mechanisms).

4. Threshold derivation and ROC presentation

Although thresholds for positivity are defined using ROC analyses, these are relegated to Supplementary Figure EV6. Given the central role these thresholds play in classifier interpretation and subsequent performance comparisons, it would be more appropriate to present the ROC curves in the main figures. This would improve transparency and help readers assess the diagnostic value of each modality.

Following the reviewer recommendation, we have moved the ROC analyses to the main Figures. The revised ROC analysis will be available in Figure 4 of the revised manuscript.

5. Selection criteria for illustrative patients

The four exemplar patients in Figure 4A-D are useful, but the rationale for their selection should be stated. Do they reflect typical, best-case, or edge-case scenarios?

All cases are shown in Fig EV5. We have tried to capture patients who represent various cases, including absence of relapse and occurrence of relapsed disease. The rationale for inclusion of each of these exemplar patients is described in the results section 2.3: "These were: Patient 1, whose disease did not relapse and responded to first-line treatment with all modalities supporting this. Patient 5, who responded to first-line treatment but later relapsed with all modalities detected at diagnosis, and all except ichorTF detected at relapse. Patient 8, who did not respond to first-line therapy and relapsed, with p100-150 bp potentially supporting this but no signal observed in other modalities at first-line treatment. Patient 15 who responded to treatment of relapsed disease with all modalities supporting this."

6. Lead-time calculation. The authors state that the lead time between molecular and clinical relapse could not be estimated due to the retrospective design and inconsistent sampling. However, Figure 4A-D clearly shows representative cases with positive cfDNA signals prior to radiological relapse. Even if a cohort-wide estimation is not feasible, the authors should report individual lead times (in days or

weeks) for these patients and comment on the potential clinical implications. A median and range for those with available data would provide valuable insight into the potential of cfDNA assays for anticipatory intervention.

We thank the reviewer for this constructive suggestion. Unfortunately, the lead time cannot be calculated due to the sampling schedule. Plasma samples were collected on treatment at clinical relapse but not before (e.g. every 3 months after baseline), so such time cannot be calculated. We are aware this is a limitation of our study that future studies should address.

Minor comments:

1. Incomplete Visualisation in Supplementary Figure EV5

Figure EV5 is a key supplementary figure supporting longitudinal analyses. However, in the axis labels, numerical tick marks are overlapping, this should be improved. since limits interpretability.

Thank for you this comment. We have modified the revised Figure EV8 to increase visibility. A summary of the results is available in revised Figure 5B.

Referee #3 (Comments on Novelty/Model System for Author):

Technical quality: multiple testing correction is missing, authors should report on cfDNA quality control (fragment analysis).

Novelty: multivariate analysis of cfDNA is novel, however in the current manuscript actual integration of the different data layers is not performed.

Medical impact: this is a proof-of-concept study with a limited number of cases/samples and therefore not clear yet how impactful this approach will be on patients.

Referee #3 (Remarks for Author):

In this manuscript the authors present a proof-of-concept study to use cfDNA methylomics and fragmentomics for relapse monitoring in pediatric sarcoma, including Ewing Sarcoma and CIC-rearranged sarcoma. Both omics layers are generated using T7-MBD-seq and analysed using ichorCNA (for ichorTF), LOLA for size distribution analysis of cfDNA, LIQUORICE at the EWS-FLI1 binding regions as well as methylation based classification. These different blood-based analytes show promise for monitoring patients with EwS or CIC and detection of relapse events, however larger patient cohorts are needed to prove clinical utility and to allow better evaluation of the multi-analyte approach for disease monitoring, which is my biggest concern and comment on the current study.

We thank the Reviewer for their time and constructive comments. We have detailed replies to their comments below.

1. Overview Figure 1 should also mention the 78 other NCCs (non-cancer controls) that are used for training, and also the in silico mixtures for validation (from Patrizi et al, 2024). In addition, the results from the latter validation cohort (in Fig EV4A) are

important to my opinion and therefore I would advise to include this figure in the main manuscript.

We have included 19 additional plasma samples from 8 patients in our study. Also, we have included an additional 17 plasma samples from NCCs and excluded 17 NCCs that were greater than 70 years old. In total we have now samples from 107 NCCs. This is reflected in the revised Figure 1, which is also including the additional NCCs not previously mentioned in this Figure. The results of the full in silico dilution results can now be viewed in Figure 3B in the revised manuscript. Due to space constraints, these are not described in Figure 1 but are shown in Figure 3A.

2. In paragraph 2.2, different analytes (eg ichorTF, LIQUORICE SCORE, etc) are tested in different comparisons, requiring multiple testing correction which seems not to be done.

We thank the reviewer for noticing the omission. In the revised version of the manuscript, we have performed Benjamini-Hochberg multiple testing correction and have included the following description in the corresponding section: "All statistical tests performed are detailed in the text and figures. Where a Mann-Whitney U test was performed, wilcox_test was used with Benjamini-Hochberg multiple testing correction (rstatix, v0.7.2 (Kassambara, 2019)) to calculate and add the p-value to each plot. Non-significant p-values are not plotted. Where detail was not provided, all tests were 2-sided with a threshold of $p < 0.05$."

3. At least a few (almost) age-matched NCCs should be included in this study to validate the real performance of the classifier/analytes.

We included 17 additional NCCs in the revised manuscript and removed 17 NCCs that were aged ≥ 70 years, and we have now in total 107 NCCs. Out of the 107 NCCs included in total, 20 are below 45 years old.

4. On the DELFI analysis: why is chromosome 8q mentioned here? What is the rationale to pick this one? What about other regions?

These regions were the most enriched in short fragments, which is matching large chromosomal gains previously reported in the context of EwS. The region was not "selected" but highlighted by the analysis of DELFI_FTK. We developed further our explanation in the corresponding result section: "Region-based cfDNA fragmentation analysis was also performed to gain a deeper understanding of the observed global fragmentation shifts. To mitigate potential sequencing coverage-based confounding effects, coverage was normalised by down sampling both NCC and cancer WGS samples to $\sim 1\times$. Subsequently, an approach similar to DELFI (Cristiano et al, 2019) was used to assess the ratio of short to long fragments across the genome in 5 Mb bins (DELFI_FTK) (Li et al, 2024). This analysis suggested an enrichment in short fragments across chromosome arms 8p and 8q, which commonly have copy number gains in EwS (Tirode et al, 2014; Fig 2A), in diagnosis and relapse samples compared to first-line treatment (Fig EV6A). Additionally, unsupervised clustering of DELFI_FTK analysis showed separation of 9/16 diagnosis and 12/18 relapse events from NCCs (Fig EV6B)."

5. Both for the LIQUORICE and the EwingSign score, it is not clear how the cut-off of -105.6 and 50%, respectively, is determined. Was this done in a completely unbiased way? Authors should elaborate more on this.

The threshold calculation for LIQUORICE and EwingSign are defined in the Methods section of the revised manuscript. They read as follows: "The threshold of positivity was set equal to the lower bound of the 95% prediction interval of the 24 tested NCCs from NCC cohort 2, which was calculated with LIQUORICE's summary tool using the default settings (Peneder et al, 2022).". For the EwingSign: "This threshold ensures that the predicted probability for the assigned class is greater than combined probabilities of other classes, which together sum to one."

approach, integrating the 4 different analytes is missing in this study. The performance of the different analytes is described separately, however the manuscript would benefit from a more integrated analysis (both in 2.2 and 2.3). As it will probably be difficult to properly perform this with the current small sample size, authors should consider to increase sample numbers, eg by involving other centers.

We agree that the limited cohort size is restricting our capacity to drive significant clinical conclusions, notably regarding the integration of different modalities to enhance performance. This is why we restricted ourselves to not perform machine learning integration due to the risk of overfitting integrative models and preferred to directly compare modalities. The research team is in contact with other centres to perform joint analysis and validation, but this will be outside the scope of the current study. Nevertheless, we have included in the revised manuscript additional plasma samples from patients with EwS (19 samples from 8 patients) recruited at the Royal Manchester Children's Hospital. We have also included 17 additional NCCs.

7. One of the last conclusions from the results is that "at least one modality was positive in 17/18 relapse samples": however it was not mentioned what was the result for non-relapse cases and for NCCs? This should also be described.

For the hold-out NCC test set (NCC cohort 2), 1/24 sample was detected positive for the P100-150bp modality, and no NCC samples were detected positive for the other modalities. For the 17 cases where the patients did not relapse (during the study timeframe), 4 were detected positive by one or more modality (Fig 5B and Fig EV8). These were Patient 9 (p100-150 bp), Patient 11 (LIQUORICE and EwingSign), Patient 16 (EwingSign) and Patient 24 (LIQUORICE). As Patient 24 was collected less than 2 years ago, it is possible that this patient may relapse in the future.

8. The performance of the p100-150 measures seems to be a bit more doubtful, with false positive detection and the least performance in the different study parts. Discussion on this is missing in the manuscript (cfr Sup Table 7).

Our previous analysis of the cfDNA size distribution illustrated by the P100-150 metric suffered from the relatively low coverage of the WGS fraction of the T7-MBD Seq. We decided to investigate the cfDNA size distribution of the methylated fraction of the method where the coverage is much higher and uniform between samples. Using this subset of the data, we observed that the cfDNA fragment size distribution can be calculated in a similar manner than with WGS (revised Figure 2C and revised Figure EV5) leading to a higher overall performance and specificity compared to the WGS fraction (revised Figure 2D and revised Figure 4A). We have amended both

our method, discussion and result sections and the corresponding figure legends to reflect this modification in our method to calculate the P100-150 metric.

9. Figure 4: the crosses indicating whether a patient died or not, are missing in the figure

The crosses indicating whether a patient died or not have been included in the revised Figure 4.

10. QC (Tapestation) of the cfDNAs should be reported. Where only good quality samples included or also samples with a lot of high molecular weight DNA?

We have now included TapeStation data reported as % cfDNA in FigEV1B. 3/90 samples have high molecular weight DNA detected with TapeStation. These three samples correspond to Ewing first-line treatment timepoints. The EwingSign classifier identified one sample as Ewing and the remaining two as NCC.

12th Jan 2026

Dear Dr. Mouliere, Dear Florent,

Thank you for submitting your revised study, and please accept my apologies for the delay in getting back to you as one referee needed more time in this busy period of the year. We have now received the reports from the three referees. As you will see below, they are overall satisfied with the revisions pending minor revisions, and I will therefore be able to accept your manuscript once the following concerns are addressed:

1/ Referees' concerns:

Please address the remaining concerns from the referees, and provide a point-by-point rebuttal letter to their comments.

2/ Manuscript text:

- Please indicate in track changes mode any new modification.
- Please remove the "Resource Availability" and "Materials Availability" sections.
- Correct the order and the headings of the sections in the manuscript text to: Abstract / Keywords / The Paper Explained / Introduction / Results / Discussion / Methods / Data Availability / Acknowledgements / Disclosure and Competing Interests Statement / References / Figure Legends / Expanded View Figure Legends
- Materials and Methods should be Methods.
- Include a statement confirming that the experiments conformed to the principles set out in the WMA Declaration of Helsinki and the Department of Health and Human Services Belmont Report.
- BioRender: remove the reference from the figure legend and add a dedicated "Graphics" section to the Methods using this format:

Graphics:

(some of the... OR Figure #... OR synopsis) Graphics were created with BioRender.com.

- Data Availability: please update the URL to the data deposited in EGA. For code access in zenodo, please remove "upon request", as per our policy all data must be made accessible to the public.
- Funding: a few funders are mentioned in the Acknowledgements and have been added to the Comments in eJP. Please remove them from the Comments and if they are important funders, please add them to the funders list; if they should only be acknowledged but did not contribute funding that should be listed, it is fine to have them in the Acknowledgments only.
- "Bibliography" heading should be changed to "References"

3/ Figures:

- Figures should be called out in chronological order (currently, Fig. EV3A is called out before Figure EV2B).
- Please upload the 7 EV tables as separate files and correct the titles in the table legends to Table EV1 etc.
- Please address the queries from our data editors in the figure legends:
 1. Please note that the legends for figure 2 is not provided in the sequential manner (legend for figure 2d,f is provided before legend of figure 2c,e). This needs to be rectified.
 2. Please note that information related to n is missing in the legends of figures EV-2A-C; EV-3B,C.

4/ I introduced minor changes in your synopsis, please let me know if you agree or amend as you see fit:

Diagnosis and monitoring of relapse in patients with Ewing sarcoma (EwS) is challenging using liquid biopsies. Current methods focus on the analysis of individual liquid biopsy modalities. A sensitive approach has been designed that integrates the methylome and fragmentome of cell-free DNA (cfDNA).

- 87 plasma samples from 23 patients with EwS, 3 CIC-rearranged sarcoma and 107 controls were sequenced using T7-MBD seq; 4 modalities were calculated.
- A machine learning model (EwingSign) was developed to detect EwS and CIC-rearranged sarcoma from cfDNA methylome profiles.
- EwingSign identified 15/18 relapse events and 11/16 diagnostic samples, with 0/24 false positives in non-cancer controls.
- When combined with global and regional fragmentome analysis, all 18 relapse cases were detected, with 15/18 detected by 2 or more modalities.
- Combined methylome and fragmentome profiling enhanced sensitivity for EwS detection and relapse monitoring in serial plasma samples.

5/ As part of the EMBO Publications transparent editorial process initiative (see our Editorial at

<http://embomolmed.embopress.org/content/2/9/329>), EMBO Molecular Medicine will publish online a Review Process File (RPF) to accompany accepted manuscripts.

This file will be published in conjunction with your paper and will include the anonymous referee reports, your point-by-point response and all pertinent correspondence relating to the manuscript. Let us know whether you agree with the publication of the

RPF and as here, if you want to remove or not any figures from it prior to publication. Please note that the Authors checklist will be published at the end of the RPF.

I look forward to receiving your revised manuscript.

With kind regards,

Lise Roth

***** Reviewer's comments *****

Referee #1 (Comments on Novelty/Model System for Author):

The technical quality, novelty, medical impact and adequacy of the model system remain unchanged from my initial review.

Referee #1 (Remarks for Author):

The authors have done a good job addressing my comments. However, a few additional details remain to be addressed:

Why are only 5 NCC samples processed through LIQUORICE? What happened to the other 24?

In the previous version of our manuscript, the majority of our NCCs were sequenced to low coverage (~0.1X). LIQUORICE, LOLA and DELFI_FTK analysis all require higher coverage (> 1.5X based on internal benchmarking). Therefore, we used only the 5 NCCs where higher coverage WGS was available for this analysis. This has been modified in the revised manuscript and sequenced deeper the 24 NCCs included in the LIQUORICE analysis.

ADDITIONAL COMMENT: Can the authors specify this in the manuscript? Ideally with the internal benchmarking data.

How do the EwigSign scores correlate with tumor fractions? If the classifier is built on tumor-specific DMRs, I would expect some correlation. I see this was done in Fig. EV4 but the authors should also compare to their ichorCNA TFs.

A direct correlation of the EwigSign prediction score and ichorTF is not appropriate to perform, as ichorTF is a quantitative metric and EwigSign a mean probability. We can however compare the 2 metrics (see Figure below for reviewers only). We can observe that the EwigSign prediction is high when ichorTF is higher than 3%. But also, that 19 samples are correctly detected by EwigSign when their ichorTF is below the limit of detection.

ADDITIONAL COMMENT 1: I do not follow why the authors believe it is not appropriate to compare ichorTF (quantitative metric) and EwigSign (mean probability) if they compared EwS class prediction scores to in silico mixtures of EwS methylation data (quantitative metric). Can't they do the same analysis comparing EwS class prediction scores to ichorCNA tumor fractions?

ADDITIONAL COMMENT 2: The authors show some compelling data that should be added to the manuscript. Why is it reviewer only?

Congrats on the beautiful work!

Referee #2 (Comments on Novelty/Model System for Author):

Technical quality is high, with a robust integrated cfDNA workflow combining orthogonal signals and clear performance reporting, though additional sensitivity analyses (notably age-restricted controls and inter-modality agreement) would further strengthen the conclusions. Novelty is high because it leverages a multimodal methylome/fragmentome strategy-including EwS-

specific TFBS-informed profiling-for EwS/CIC monitoring, where mutation-based ctDNA assays are often limited. Medical impact is medium: the approach is clinically promising for non-invasive relapse monitoring, but current evidence is pilot/retrospective and requires larger prospective validation with standardized sampling and lead-time assessment. The model system is adequate, based on longitudinal human plasma cfDNA; ethical issues are limited to standard human biospecimen considerations (consent, privacy, governance).

Referee #2 (Remarks for Author):

I have reviewed the revised manuscript and the authors' point-by-point responses. Overall, the revision is improved. The study remains timely and potentially impactful for relapse monitoring in EwS/CIC using cfDNA methylome/fragmentome-derived signals. However, a small number of key issues still require clarification/analysis to avoid overinterpretation and to strengthen the main claims.

Major comments

1) Residual age mismatch and potential confounding

I appreciate the expansion of non-cancer controls (NCC) and the added discussion acknowledging the age imbalance and its association with fragmentation metrics. However, the current revision still does not fully resolve whether age-related effects could influence the reported specificity and/or decision thresholds, particularly for EwingSign (methylation-based) in addition to p100-150.

Requested additions:

- Provide a sensitivity analysis restricted to younger controls (e.g., NCC <45 years; the authors mention a subset is available). Report the distribution of EwingSign scores (and/or predicted class probabilities) and the number of false positives at the chosen threshold.
- Add a simple correlation analysis in NCC: age vs (i) p100-150 and (ii) EwingSign score/probability (Spearman is fine), in Supplementary.
- Ensure the Discussion explicitly addresses the possibility of age effects on methylome-based classification, not only fragment length.

This is important because the key message relies on high specificity in NCC; demonstrating robustness to age composition would substantially strengthen the manuscript.

2) Lead time claims: revise wording and/or report per-patient lead time where possible

The authors state that lead time cannot be estimated due to sampling occurring at clinical relapse and not before. However, the longitudinal descriptions/plots appear to include timepoints prior to clinical relapse (at least in some patients), with rising cfDNA signals preceding the clinically documented relapse.

Requested change:

- Where pre-relapse samples exist, provide a per-patient lead time table (Supplementary is sufficient): date/day of clinical relapse, date/day of first positivity for each modality, and the interval (days).
- Avoid wording that could be interpreted as "no pre-relapse sampling existed" if the figures/case narratives indicate otherwise.

Referee #3 (Comments on Novelty/Model System for Author):

Compared to the first submission, this manuscript has significantly improved, however due to the pilot phase of this study, the medical impact is currently low, but the study presents important POC data for the research community.

Referee #3 (Remarks for Author):

A few small remarks remain:

Line 39 in abstract: also mention the number of false positives among the NCC for the combined model.

Line 81: CIC is not properly introduced: why were also CIC patients included and not only focus on EwS

Line 135: explain the choice of cut-off 0.03

Line 140, Fig2C: In the figure add a raster, so that it is more easy to see that there is a difference in size distribution among the 3 groups.

Line 225: talk also about the 5 diagnostic samples that were never positive in the combined analysis. Is there an explanation? E.g. in discussion.

****** Reviewer's comments ******

Referee #1 (Comments on Novelty/Model System for Author):

The technical quality, novelty, medical impact and adequacy of the model system remain unchanged from my initial review.

We thank the reviewer for their comments.

Referee #1 (Remarks for Author):

The authors have done a good job addressing my comments. However, a few additional details remain to be addressed:

Why are only 5 NCC samples processed through LIQUORICE? What happened to the other 24?

In the previous version of our manuscript, the majority of our NCCs were sequenced to low coverage (~0.1X). LIQUORICE, LOLA and DELFI_FTK analysis all require higher coverage (> 1.5X based on internal benchmarking). Therefore, we used only the 5 NCCs where higher coverage WGS was available for this analysis. This has been modified in the revised manuscript and sequenced deeper the 24 NCCs included in the LIQUORICE analysis.

ADDITIONAL COMMENT: Can the authors specify this in the manuscript? Ideally with the internal benchmarking data.

We have included this explanation in the new revision of our manuscript (in the section 4.10 of the Methods). The new paragraph reads as follow: "Due to the requirement of the employed region-based fragmentomic methods of higher sequencing depth (Cristiano et al, 2019; Peneder et al, 2022, 2021) and increased sensitivity to coverage differences, only the higher depth sequencing of the 24 WGS cohort 2 NCCs was used, with the cfDNA WGS NCC and cancer samples down sampled to an average sequencing depth of 1x (0.3-1.2-fold coverage).

How do the EwigSign scores correlate with tumor fractions? If the classifier is built on tumor-specific DMRs, I would expect some correlation. I see this was done in Fig. EV4 but the authors should also compare to their ichorCNA TFs. A direct correlation of the EwigSign prediction score and ichorTF is not appropriate to perform, as ichorTF is a quantitative metric and EwigSign a mean probability. We can however compare the 2 metrics (see Figure below for reviewers only). We can observe that the EwigSign prediction is high when ichorTF is higher than 3%. But also, that 19 samples are correctly detected by EwigSign when their ichorTF is below the limit of detection.

ADDITIONAL COMMENT 1: I do not follow why the authors believe it is not appropriate to compare ichorTF (quantitative metric) and EwigSign (mean probability) if they compared EwS class prediction scores to in silico mixtures of EwS methylation data (quantitative metric). Can't they do the same analysis comparing EwS class prediction scores to ichorCNA tumor fractions? ADDITIONAL COMMENT 2: The authors show some compelling data that should be added to the manuscript. Why is it reviewer only?

A direct correlation between the values is challenging due to the non-quantitative nature of the EwigSign score, and the requirement for supervised training (at the difference of ichorCNA). We were concerned this would be interpreted as an unfair comparison. Based on the reviewer suggestion we have now included this comparison of the ichorTF and EwigSign score (previously only available as a Figure for reviewers, see **Figure EV8**).

Congrats on the beautiful work!

Thank you!

Referee #2 (Comments on Novelty/Model System for Author):

Technical quality is high, with a robust integrated cfDNA workflow combining orthogonal signals and clear performance reporting, though additional sensitivity analyses (notably age-restricted controls and inter-modality agreement) would further strengthen the conclusions. Novelty is high because it leverages a multimodal methylome/fragmentome strategy—including EwS-specific TFBS-informed profiling-for EwS/CIC monitoring, where mutation-based ctDNA assays are often limited. Medical impact is medium: the approach is clinically promising for non-invasive relapse monitoring, but current evidence is pilot/retrospective and requires larger prospective validation with standardized sampling and lead-time assessment. The model system is adequate, based on longitudinal human plasma cfDNA; ethical issues are limited to standard human biospecimen considerations (consent, privacy, governance).

We thank the reviewer for their comments.

Referee #2 (Remarks for Author):

I have reviewed the revised manuscript and the authors' point-by-point responses. Overall, the revision is improved. The study remains timely and potentially impactful for relapse monitoring in EwS/CIC using cfDNA methylome/fragmentome-derived signals. However, a small number of key issues still require clarification/analysis to avoid overinterpretation and to strengthen the main claims.

Major comments

1) Residual age mismatch and potential confounding. I appreciate the expansion of non-cancer controls (NCC) and the added discussion acknowledging the age imbalance and its association with fragmentation metrics. However, the current revision still does not fully resolve whether age-related effects could influence the reported specificity and/or decision thresholds, particularly for EwingSign (methylation-based) in addition to p100-150.

Requested additions:

- Provide a sensitivity analysis restricted to younger controls (e.g., NCC <45 years; the authors mention a subset is available). Report the distribution of EwingSign scores (and/or predicted class probabilities) and the number of false positives at the chosen threshold.
- Add a simple correlation analysis in NCC: age vs (i) p100-150 and (ii) EwingSign score/probability (Spearman is fine), in Supplementary.

- Ensure the Discussion explicitly addresses the possibility of age effects on methylome-based classification, not only fragment length.

This is important because the key message relies on high specificity in NCC; demonstrating robustness to age composition would substantially strengthen the manuscript.

As suggested, we added age-related analyses for the p100–150 and EwingSign modalities by plotting the distribution of p100–150 and EwingSign cancer prediction scores for NCCs across different age groups. These results are shown in Figure EV10, and possible age effects are discussed in the Discussion section. This addition reads as follows: “While across all age groups NCCs showed significantly lower p100-150 bp than cancer patients (Fig EV10A) and only a single age-matched NCC (< 45 years) was assigned a cancer prediction score above the classification threshold by EwingSign (Fig EV10B), further work is needed to fully examine the effects of age on EwS fragmentomic and methylomic analyses.”

For the correlation analysis, age is only available in categorical form (Table EV3) rather than as a continuous variable, which limits our ability to perform a formal correlation analysis of the age with these measures.

2) Lead time claims: revise wording and/or report per-patient lead time where possible. The authors state that lead time cannot be estimated due to sampling occurring at clinical relapse and not before. However, the longitudinal descriptions/plots appear to include timepoints prior to clinical relapse (at least in some patients), with rising cfDNA signals preceding the clinically documented relapse.

Requested change:

- Where pre-relapse samples exist, provide a per-patient lead time table (Supplementary is sufficient): date/day of clinical relapse, date/day of first positivity for each modality, and the interval (days).

- Avoid wording that could be interpreted as "no pre-relapse sampling existed" if the figures/case narratives indicate otherwise.

Only 3 patients with clinical relapse (patients 8, 12, 5) have at least a pre-relapse plasma sample to evaluate lead-time. Among them all 3 of them have at least one cfDNA modality positive at clinical relapse. Patient 13 has no cfDNA sample collected between baseline and relapse (both are positives). Patient 8 has two sample collected on-treatment before relapse, one of them being positive for one modality. Patient 5 has two cfDNA samples collected before the first clinical relapse, one of them being positive for 2 modalities. The potential lead-on time is 119 days for patient 8, and 351 days for patient 5.

Due to the small number of patients with clinical relapse and a pre-relapse plasma sample available, we have not included a new supplementary table. We have included the description above in the corresponding section of the results. We however do not think we can claim such lead time observation viewing the small number of samples supporting the information. We have altered the wording in the discussion to reduce potential confusion regarding the absence of pre-relapse samples.

Referee #3 (Comments on Novelty/Model System for Author):

Compared to the first submission, this manuscript has significantly improved, however due to the pilot phase of this study, the medical impact is currently low, but the study presents important POC data for the research community.

We thank the reviewer for their comments.

Referee #3 (Remarks for Author):

A few small remarks remain:

Line 39 in abstract: also mention the number of false positives among the NCC for the combined model.

This is amended in the revised manuscript. The corresponding sentence reads as follows: "When combined with global and regional fragmentome analysis, all 18 relapse cases were detected, with 15/18 detected by 2 or more modalities, and 1 out of 24 NCC was detected by one modality."

Line 81: CIC is not properly introduced: why were also CIC patients included and not only focus on EwS

We aimed to collect samples from all patients with Ewing sarcoma and other 'Ewing-like' small round cell tumours driven by gene fusions (CIC::DUC4, BCOR::CCNB3, etc). No participants were recruited with other gene fusions. The corresponding sentence in the revised manuscript now reads as follows: "This pilot study aims to longitudinally profile cfDNA samples from patients with EwS and Capicua-rearranged sarcoma (CIC), two cancers with small round cell tumours driven by gene fusions, using the in-house T7-MBD-seq assay, an enrichment-based methylation capture method using a methyl binding domain protein (Conway et al, 2024; Chemi et al, 2022)."

Line 135: explain the choice of cut-off 0.03

The cut-off of 0.03 is recommended by the authors of the ichorCNA metric (PMID: 29109393). We included a slight addition to this line which now reads as follows: "Using copy number aberrations (Fig 2A), ctDNA was detectable in 9/16 diagnosis samples, 0/32 first-line treatment samples, 10/18 samples at disease relapse and 4/20 samples taken on treatment after relapse (cut-off of 0.03 ichorTF as advised by its authors (Adalsteinsson et al, 2017), **Fig 2B, Fig EV2A**)."

Line 140, Fig2C: In the figure add a raster, so that it is more easy to see that there is a difference in size distribution among the 3 groups.

We modified Figure EV3 to include a comparison between the 3 groups as suggested. We decided to use a log2ration of the size distribution comparing EwS to NCC, CIC to NCC, and EwS to CIC to clarify the differences.

Line 225: talk also about the 5 diagnostic samples that were never positive in the combined analysis. Is there an explanation? E.g. in discussion.

We have added discussion of the 5 negative diagnostic samples in the discussion section of the manuscript. This addition reads as follows: "Disease was detected at diagnosis in 11/16 patients using EwingSign. In the five patients where no disease was detectable by liquid biopsy, patients all had localised disease."

29th Jan 2026

Dear Dr. Mouliere, Dear Florent,

Thank you for submitting the revised files. I am pleased to inform you that your manuscript is accepted for publication and is now being sent to our publisher to be included in the next available issue of EMBO Molecular Medicine.

You may qualify for financial assistance for your publication charges - either via a Springer Nature fully open access agreement or an EMBO initiative. Check your eligibility: <https://link.springer.com/journal/44321/how-to-publish-with-us>

With kind regards,

Lise

>>> Please note that it is EMBO Molecular Medicine policy for the transcript of the editorial process (containing referee reports and your response letter) to be published as an online supplement to each paper. If you do NOT want this, you will need to inform the Editorial Office via email immediately. More information is available here: <https://link.springer.com/partners/embo-press/editorial-policies#Peer%20review>